# Artificial intelligence empowering museum space layout design: Insights from China

**Qiang Tang[1], Liang Zheng[2]\*, Yile Chen[2]\*, Lina Yan[2]\*, Junzhang Chen[3]**

**1** School of Design, Shunde Polytechnic, Shunde District, Foshan City, Guangdong Province, China,
**2** Faculty of Humanities and Arts, Macau University of Science and Technology, Taipa, Macau, China,
**3** Faculty of Innovation and Design, City University of Macau, Taipa, Macau, China

\* 2009853gat30002@student.must.edu.mo (LZ); 2009853gat30001@student.must.edu.mo (YC); 2009853SAT30001@student.must.edu.mo (LY)

**Data Availability Statement:** There are ethical restrictions which prevent the public sharing of minimal data for this study. Data are available from Dr. Linhui Hu, Guangdong University of Technology's School of Art and Design, via email

## Abstract

The floor plan layout of museum exhibition spaces is the skeleton network of the museum, which determines the internal circulation and spatial form of the museum. This paper studies the method and practice of using artificial intelligence technology to assist in the space design of exhibition halls in urban cultural museums. First, it introduces the limitations of traditional space design methods for exhibition halls in urban cultural museums and the superiority and application prospects of the CGAN (conditional generative adversarial network) model in space design. Second, the principle and training process of the CGAN model are explained in detail, and the experimental results and analysis are given. By learning 100 floor plans of exhibition halls of urban culture museums, the CGAN model can generate a new floor plan design for an exhibition hall, which provides a new idea and innovative method for this design task. Finally, the limitations and future research directions of the CGAN model in the space design of urban cultural museum exhibition halls are discussed. The study shows that using the CGAN model to learn the floor plans of exhibition halls of urban cultural museums can effectively improve the innovation and practicability of space design and has the following advantages: (1) It can quickly generate a large number of exhibition hall floor plans, shorten the design cycle, and improve design efficiency. (2) The generated floor plan designs of the exhibition hall are diverse and personalized, meeting the design requirements of different scenarios and needs. (3) The method promotes the deep integration of space design and artificial intelligence technology and provides new possibilities and ideas for space design. These conclusions provide new ideas and methods for the space design of exhibition halls of urban cultural museums and provide a reference and inspiration for space design and intelligent applications in other fields, such as office space design, home decoration space design, landscape space design, and historical arcade and building renovation design.

(hlh@gdut.edu.cn), for researchers who meet the criteria for access to confidential data, after receiving approval from the Institutional Review Board (IRB) of Shunde Polytechnic representative, Dr. Long Gan, via email (10089@sdpt.edu.cn). The dataset of training images for this study are publicly available from the Mendeley Data repository (https://data.mendeley.com/datasets/9bzpjyh7ft/1).

**Funding:** This study was supported by the research from the Guangdong Provincial Department of Education's key scientific research platforms and projects for general universities in 2023: Guangdong, Hong Kong, and Macao Cultural Heritage Protection and Innovation Design Team (Funding Project Number: 2023WCXTD042; Shunde Polytechnic Internal Number: 2023-KJZX047); 2020 Guangdong Province Ordinary Universities Characteristic Innovation Project (No. 2020WTSCX286); 2019 Guangdong Province Ordinary University Philosophy and Social Science Project (No. 2019GXJK131); Shunde Polytechnic "Tang Qiang Cantonese Area Cultural Heritage Protection Design Skills Master Studio".

**Competing interests:** The authors have declared that no competing interests exist.

# 1. Introduction

## 1.1. Research background: The digital construction trend in museums

Since 2020, museums have become a strategic part of the United Nations Sustainable Development Goals (SDGs). They are key players in promoting physical and mental health and sustainable community development [1]. Museums can have positive effects through research, education, exhibitions, and community participation. After 2000, in China, there was an upsurge in the building of museums, which further promoted domestic museum development [2, 3]. With the development of society and the economy, people's spiritual and cultural needs tend to become diversified, and museums have gradually become cultural places for daily study and leisure visits [4, 5]. In 2023, relevant departments noted, when discussing the construction of a cultural heritage protection, inheritance, and utilization system during the "China 14th Five-Year Plan" period, that it was important to "Stimulate the vitality of museums and comprehensively promote the high-quality development of museums" [6, 7]. It is also emphasized here that the vitality of museums and other cultural relic protection facilities should be supported, the level of cultural relic display should be improved, and "going to museums to see exhibitions and receive education" should be made a way of life [8, 9]. In this context, digital technology is being used increasingly widely in museums, such as in experience pavilions for VR devices, holographic projections, and light and shadow performances [10]. However, this pertains primarily to the digitization of museum exhibit content. Incorporating cutting-edge technology such as digitalization or artificial intelligence, particularly in the floor plan design of the core exhibition space of the museum, is a question to consider in the context of the current construction of museum buildings in response to high demand.

We have entered not only the digital age but also the era of artificial intelligence [11]. Therefore, artificial intelligence not only provides new opportunities for social research but also necessitates new research paradigms and methods for studying the current digital society [12]. Similarly, museums, as important social and cultural places, preserve the past in a way that highlights the importance of artifacts, but content is as important as form when appreciating art and history [13]. If museums want to manage an increasing number of documents in the technological age, they should introduce intelligent systems to handle large quantities of data rather than relying on curators and archivists as is traditional [14]. For example, artificial intelligence helps museums manage digital archives and files, and robots that interact with visitors help museums analyze data, improve the visitor experience, and predict future developments [15–17]. Artificial intelligence in the abovementioned fields has a common feature: it frees the labor force from automated work [14]. Museums are no exception. Based on this background, this study explores the methods and practices of integrating artificial intelligence technology into exhibition hall design for urban cultural museums.

## 1.2. Literature review

Regarding museum display layouts, some scholars believe that it is necessary to highlight the characteristics and advantages of regional history and culture, display exhibition content through scientific and reasonable layout design, and avoid "thousands of museums being the same" [18]. It is also believed that museum exhibition space layout design has the following three functions: forming spatial streamlines, establishing narrative rhythm, and creating situations [18–21]. This also means that architects need to devote considerable effort to spatial layout. A lack of explanation and consideration of exhibition content will make the layout plan very traditional, and it may even appear rigid [22–24]. Therefore, planning the exhibition hall layouts is the core of museum architectural design [25, 26]. In China, some scholars equate

museum space with the exhibition building and believe that exhibition activities in the building are carried out through the theme of the building [27, 28]. This facilitates message transmission, acceptance, and inspiration between people and between people and things. In the past, in the face of a large number of museum construction needs, the traditional method relied on the knowledge and experience of professional architects to make a series of drawings to guide construction [29]. However, this approach is prone to bias during the design process and can lead to inefficiencies. For example, when designing a museum building, the architect needs considerable time to read the background information about the museum in the early stages [30]. This information includes the museum's urban cultural background, site design background, space layout, exhibition content requirements, and exhibition hall layout [31]. Moreover, architects also need to individually conceive each space and determine how each exhibition hall should be arranged [32–35]. These tasks are repetitive in character [36, 37]. Additionally, even when designing the same type of museum exhibition hall, due to different regional characteristics, the professional experience of architects may differ, which may eventually be reflected in the project construction quality [38].

To improve architectural design efficiency and accuracy, in recent years, many scholars have proposed methods based on artificial intelligence, digitization, or parametrics to construct venues, thereby reducing the repetitive work for architects. For example, many scholars have proposed virtual museum design based on 360-degree panoramas, Web 3D technology, and 3D modeling technology and applied it to cultural heritage museums in different places, such as China, Europe, and the Middle East [39–43]. In addition, artificial intelligence-assisted architectural floor plan design, especially CGAN (conditional generative adversarial network) technology, has been employed and has attracted considerable attention. A CGAN is a machine learning model for generating synthetic images that can generate images similar to training data under given conditions [44]. The CGAN model captures the characteristics of a building through training data and generates a new floor plan design scheme on this basis, which improves the design efficiency and ensures architectural consistency [45]. For example, some scholars have proposed using agent-based modeling combined with a deep learning GAN algorithm to generate automated 2D building layouts [46], identify and generate floor plan designs for apartment buildings [47–49], and conduct elementary school campus planning [50]. In response to the single functional layout and poor living experience of homestays around the Paifang Street Scenic Spot in Chaozhou City, Guangdong, some scholars used Python to crawl comment data from travel websites to capture user needs and incorporated a CGAN to design the architectural homestay layouts [51]. From a smaller-scale perspective, some scholars have studied CGAN designs to divide indoor space functional areas [52], explored compressed data methods to improve GAN and CGAN training efficiency to generate architectural floor plan designs [53], and explored the application of several artificial intelligence algorithm models to museum exhibition layouts [54]. From the perspective of algorithm technology improvement, some scholars have also proposed new models, such as ArchiGAN, House-GAN, Roof-GAN and Wasserstein GAN, to automatically generate plane layouts in the architectural field [55–58]. However, different building types have different characteristics. At present, machine learning in the floor plan design of museum exhibition halls is still being explored.

## 1.3. Problem statement and objectives

Museum display design is an industry that traditionally relies on architects for design [24]. In the early stages of space design and layout, architects need to spend considerable effort to gradually lay out the exhibition hall floor plan according to the exhibition schedule [59]. It is

necessary to constantly experiment with different methods and try different types of layouts to adapt to the needs of the general exhibition content. After a suitable layout method is found, it needs to be corrected many times [60, 61]. However, if architects can use floor layout knowledge extracted by artificial intelligence to provide the museum with multiple sample choices, the architect can then compare them with the exhibition theme. This allows architects to find a suitable floor plan and then begin to optimize the layout of the museum exhibition hall, which saves considerable time and labor (Fig 1).

## 2. Methodology and materials

### 2.1 Research methods

This paper presents a method for the automatic generation of museum exhibition hall plans using CGANs. The conditional generative adversarial network (CGAN) is an extension of the generative adversarial network (GAN). It adds conditional variables to the input, ensuring that the generated samples not only conform to the data distribution but also meet specific conditional requirements. Compared to GAN and stable extension models, CGAN can better control the generation results, resulting in samples that meet specific requirements. Specifically, it has the following advantages: (1) Ensuring precise control over the generation process is crucial. CGAN permits the imposition of conditional constraints during the generation process. These conditions, including the outline and functional zoning of the museum floor plan, are included in this study. This conditional control ensures that the generated floor plan not only conforms to the museum's structural outline but also meets the functional zoning requirements, resulting in accurate floor plan generation. (2) Better handling of complex structures. Museum floor plans usually contain complex structures and diverse functional zoning. CGAN performs well in handling such complex structures because it can learn the complex patterns of data distribution through the adversarial training process and generate high-quality samples. In contrast, other generative models, such as variational autoencoders (VAE), may face greater challenges in handling complex structures. (3) Existing successful cases and experience. CGAN has many successful applications in image generation and image restoration. For

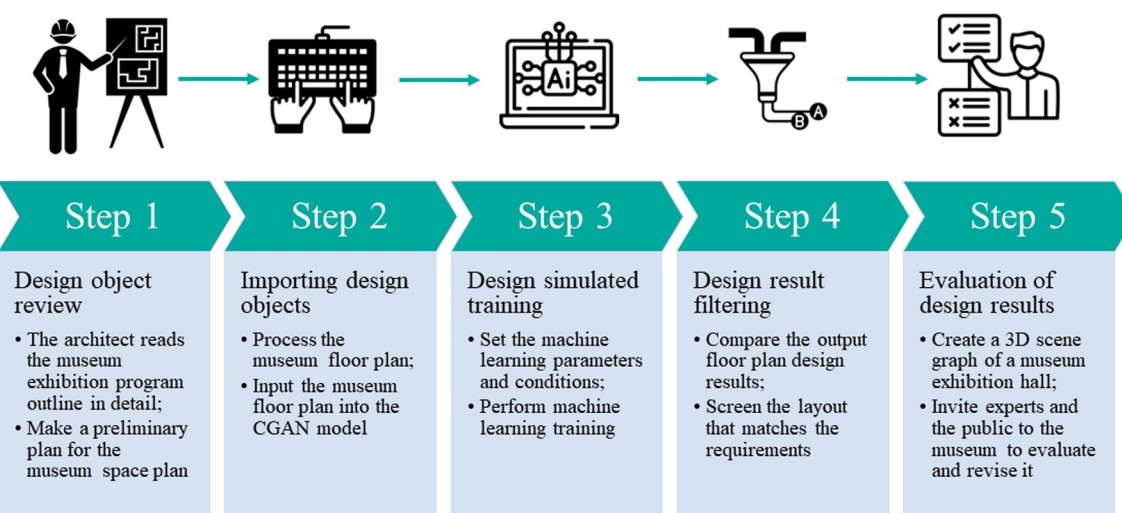

**Fig 1. Concept of artificial intelligence aiding the cultural museum design process.** This paper focuses primarily on the following three issues: (1) How can museum exhibition hall floor plan design be automated? (2) How do the results of CGAN model training meet the graphic layout design needs of a cultural museum exhibition hall? (3) How can we evaluate the generated design results of museum exhibition hall floor plans with the assistance of artificial intelligence?

example, in image restoration tasks, CGAN can generate high-quality image details significantly better than traditional methods. Therefore, choosing CGAN can draw on these successful experiences and improve the effect of museum floor plan generation.

Compared with previous studies [51, 53], the proposed CGAN model is not only optimized for museum floor plan characteristics in terms of data processing and model training. For example, the museum floor plan is divided into 10 color blocks. Each color block represents an important function in the museum floor plan so the CGAN model shows higher richness and efficiency in training and generating museum floor plans. In addition, compared with other research based on GAN model derivation, such as ArchiGAN, House-GAN, Roof-GAN, and Wasserstein GAN [55–58], the proposed CGAN method shows higher flexibility and adaptability when dealing with complex museum space layouts. For example, the proposed CGAN model can be adapted to museums with different floor plans. Therefore, the core contribution of this study is that the proposed CGAN model shows certain advantages in design process automation, efficiency improvement, and design innovation. This method helps designers better complete the design of museum exhibition halls. This intelligent design method not only simplifies the design process but can also automatically generate design solutions and improve design efficiency. In addition, this method is suitable for the space design of similar exhibition halls and has wide application value. The research method consists of five parts: data capture, data processing, model training, model evaluation, and model application (Fig 2).

1. Data capture: collect a large amount of planar data from museum exhibition halls and provide it as training data for the model. To enable model training to recognize the spatial characteristics of museum exhibition halls, this study collected nearly 60 modern museum projects that mainly display urban culture in past design work and selected 100 plans of museum exhibition hall projects. These exhibition hall project plans cover different types of museum exhibition halls with different space sizes and different layout forms. These museum exhibition hall plan data provide enough learning samples for model training that the model can obtain the exhibition hall plan layout features.

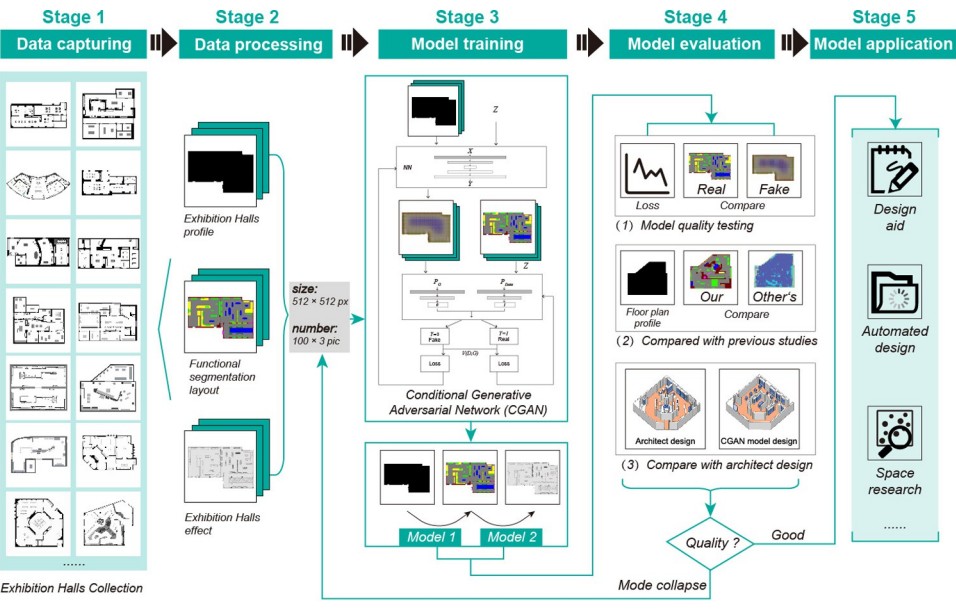

**Fig 2. Research methods and process.**

2. Data processing: classify and process the collected data to improve the effect of model training. Since the direct use of data will lead to uncontrollable situations, to avoid large deviations in model training, it is necessary to classify data pictures. According to the characteristics of the exhibition halls, the researchers classified the outlines, functional types, and floor plan features of the halls. Specific treatment methods included the following:

   a. Outlining the introduction of the exhibition hall plan. The collected floor plan outline of the exhibition hall was redrawn to form a floor plan profile (FPP).

   b. Functional classification of exhibition halls. The researchers classified each functional part of the exhibition hall by area and element (ground, showcase, background wall, etc.). The areas and elements of each function were given respective representative colors to distinguish them, forming a functional segmentation layout (FSL).

   c. Combining the plan renderings of the exhibition hall. To observe the results of the training more intuitively, the effect map of the sample floor plan and the functional segmentation layout (FSL) were sorted to form the floor plan effect (FPE).

   d. Unification of image size. The above three types of image sets were uniformly processed into images of dimensions 512×512 pixels, 96 dpi resolution, and 24-bit depth. The uniform processing of pictures can prevent errors caused by differences in sample sizes and promote the effect of model training.

3. Model training: train the model step by step to control the generation effect of the model in each step. The CGAN model framework training requires paired samples (for the training conditions for machine learning, please refer to S1 File for the operating environment). To better control the training effect, step-by-step training was conducted on the three picture sets compiled above. The specific steps were as follows:

   a. Training the control of the outline of the exhibition hall: the floor plan profile (FPP) and the functional segmentation layout (FSL) were paired for training to form Model 1.

   b. Training the intelligent layout of the internal functions of the exhibition hall: the functional segmentation layout (FSL) and the floor plan effect (FPE) were paired for training to form Model 2. After two model trainings of Models 1 and 2, the generated floor plan of the exhibition hall was finally obtained.

4. Model evaluation: evaluate the effect of model training. To verify the effect and usability of the model training results, a comparative evaluation was conducted in three parts. The steps here specifically include the following three points:

   a. Observing the loss value of each iteration in the CGAN and the test picture during model iteration. The loss value can reflect the quality of the generated samples, and the lower the loss value is, the better the model training. From the overall trend of the loss value, if the loss value continues to decrease during the training process, the generated samples are closer to the real samples, and the model training effect is better. At the same time, we observe whether the test pictures in the model iteration process gradually become more realistic and clear and whether there are blurred and distorted images.

   b. Comparison of the generation effects with those of previous models of the same type. The researchers compared the effects generated by this model with the effects of previous studies to determine whether the effect of this model is better.

c. Comparison of the effect of the actual space designed by the architect. The research evaluation process included designing a questionnaire survey for 297 participants (264 valid responses) to evaluate the public's satisfaction with the comparison between the museum floor plan generated by CGAN and the floor plan designed by humans. The survey period for questionnaire distribution starts on October 1, 2023, and ends on November 30, 2023. Five groups of exhibition hall shapes were selected for the survey. Each group included an architect's design and a CGAN-generated floor plan. Participants rated these drawings from 1 to 5. This was followed by reliability analysis using Cronbach's alpha method in SPSS21 software; validity analysis using the Kaiser–Meyer–Olkin (KMO) method and Bartlett's test of sphericity; difference analysis using the T test; and analysis of variance (ANOVA). A comprehensive evaluation was conducted through the above survey design, participant population statistics, and professional statistical methods to ensure the reliability and validity of the research results. After passing the above three evaluations, whether the effect and quality generated by this model were satisfactory was determined. If the effect was not good, further optimization cycle training was adopted to improve the training effect of the model by repeatedly adjusting the training parameters, learning rate, and network structure. When the training effect was satisfactory, the researchers proceeded with the application.

5. Model application: This method and its generated results were applied to an actual design. After training is completed, the model can automatically generate the floor plan of an exhibition hall according to the outline of the exhibition hall, which can be used as an auxiliary tool for architects to study and design the museum exhibition hall space.

## 2.2 Material collection and preprocessing

To ensure the reliability and universality of the research results, the experimental material samples in this study are from cultural museums in 63 cities in the Chinese mainland, covering museums with different geographical, cultural, and historical backgrounds (for information on the 63 museums, please refer to S2 File: Research Materials (Museum Floor Plan) Source Statistics). The sample covers museums built from 1929 to 2023. This time span covers a variety of architectural styles in China, from the early Republic of China period to modern times. According to statistics, the average area of these museums is 20,571.74 square meters, indicating that Chinese museums generally tend to have larger building scales. Additionally, the standard deviation of the area of these museums is 2,1160.37 square meters, indicating that there are significant differences in the size of each museum, reflecting the differences in economic development and the uneven distribution of cultural resources between different cities. Well-known museum design engineering companies in China (such as Guangdong Jimei Design Engineering Company, Hangzhou Zhengye Decoration Design Co., Ltd., Shenzhen Silk Road Cultural Development Co., Ltd., and Guangzhou Litian Exhibition Design Engineering Co., Ltd.) provided the majority of the collected museum exhibition hall floor plans. These floor plans are real examples of designs made by these companies in the last 5 years. In addition, a small portion of the designs were mapped by members of the research team during past practical projects involving museum design. During the sample collection process, the survey found that the graphic design of the exhibition halls of different museums has certain differences and characteristics. For example, some museums may focus more on exhibiting local cultural traditions, while others focus more on technology and modernization. Additionally, there are some general rules and commonalities. For example, to allow visitors to better understand the

exhibits, most museum exhibition halls adopt a certain streamlined layout, display area settings, and exhibit classification and labeling. The researchers classified the current types of many museum planes into rectangle, square, double arc, oval, L-shape, and trapezoid shapes. The floor plans of most Chinese museum exhibition halls take the above graphic elements as the basic shapes and then add appropriate enrichment.

It is important to note that obtaining a large amount of high-quality floor plan data is not an easy task, as collecting museum floor plans can be costly and challenging. In this study, 100 museum floor plans from 63 cities in China were collected based on existing resources and practical feasibility. At the same time, in order to make up for the problem of insufficient data volume, this paper adopts a variety of data enhancement techniques, including rotation, flipping, cropping, and scaling. These techniques can effectively expand the diversity of training data, thereby improving the generalization ability of the model. Through data enhancement, researchers can expand 100 original datasets into thousands of variants, providing the model with richer training samples. This study selected 100 representative exhibition hall plans in these museums and uniformly scaled these plans to ensure that the samples had the same size (512×512 pixels). This decision is based on the following considerations. (1) Unified image scaling to the same size can simplify the model input processing, which is crucial to maintaining the efficiency and stability of the model. (2) By unifying the image size, researchers can provide a unified data format and information density for all designs, which is important for consistent training and prediction accuracy. However, it is inevitable that (1) scaling the image to a uniform size may change the resolution of the original image, which may result in the loss or distortion of some details; (2) keeping the original resolution of the image means that the model needs to process images of very different sizes, which will greatly increase the complexity of model training and inference; And (3) images of different sizes may introduce inconsistent data representations, which may affect the model's generalization ability and prediction accuracy. In summary, the decision to unify image sizes is based on comprehensive considerations of practicality and model performance.

This study used a raster-based approach rather than a vector-based approach to process image data. This decision was based on the following considerations. (1) Raster-based methods show greater flexibility and efficiency in processing high-resolution images, especially when using deep learning and neural network models. This enables the model to capture details more accurately, resulting in higher-quality image results. (2) Although vector-based methods have unique advantages in processing graphics and design data, in the application scenario of this study, raster methods provided a more direct and effective method to train the model. (3) It should be noted that AI and machine learning technology can be flexibly applied to both raster and vector methods. In this study, the raster-based method is more suitable for generating high-quality images through deep learning. This method can more effectively utilize the capabilities of modern AI technology when dealing with complex image and pattern recognition tasks.

According to the needs of this experiment, the researchers further divided the 100 urban culture museum exhibition hall floor plans into three parts: FPPs, FSLs, and FPEs. Each section had 100 images, for a total of 300 images (Fig 3). The FPP is a simplified surface representation of the exhibition hall plan, and its color is black (R0, G0, B0); it is used to describe the shape and structure of the exhibition hall. The FSL is the result of subdividing the exhibition hall plan; it divides the exhibition hall into different areas and elements and is used to plan exhibits and tour routes. In this study, color is the classification label (please refer to S3 File for the RGB color values used in making labels). The purpose is to allow the model to distinguish the meanings of different colors so that it can generate the desired results. The type of label was determined after discussions with professionals in museum exhibition halls. They believe that display cabinets of different shapes have a great impact on the exhibition hall design. For

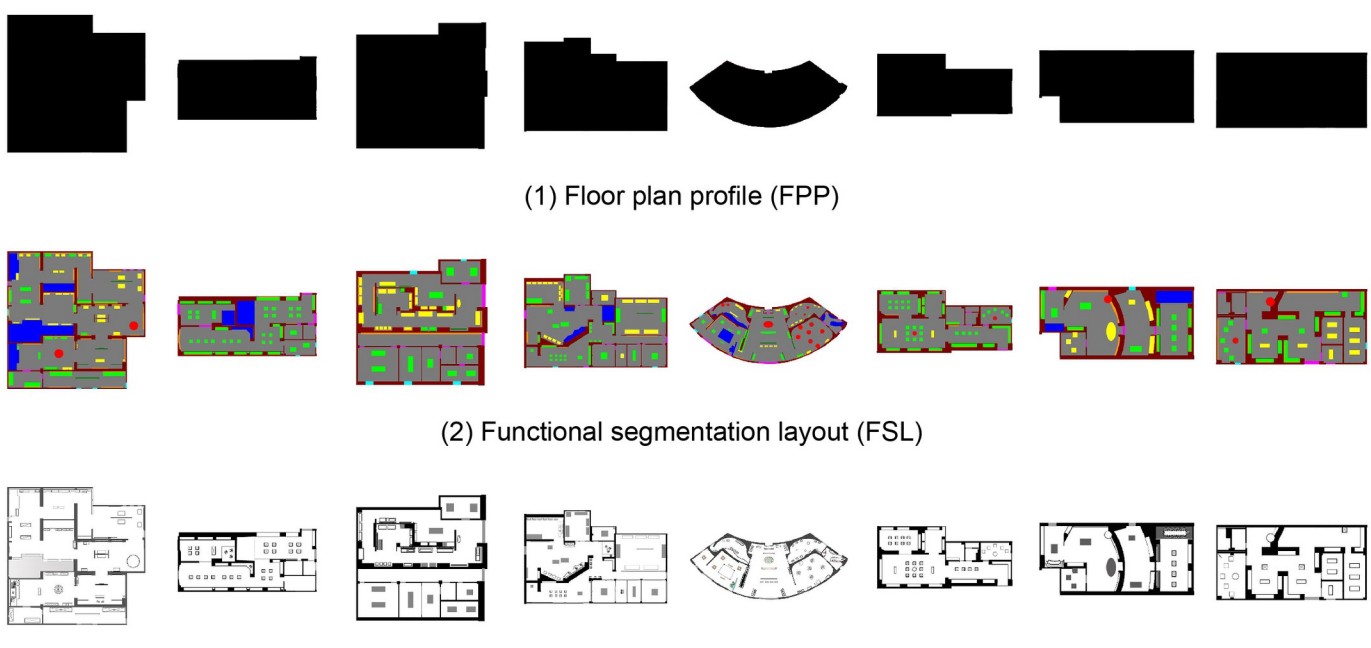

**Fig 3.** Materials used in the research: (1) floor plan profile (FPP); (2) functional segmentation layout (FSL), and (3) floor plan effect (FPE).

example, circular display cabinets are more suitable for arranging exhibits that require 360˚ viewing. In addition, there are many methods for dividing functional space and dividing space according to spaces such as display and transportation corridors. Before this study was carried out, the result of discussions with experts in related fields was that it was more practical to use indoor furniture such as display cabinets and booths as dividers. Therefore, this article chose the current method, and future researchers can also try different spatial division methods to conduct research. The FPE transforms the exhibition hall plan into a specific plan rendering, including the seat layout, display layout, and colors commonly used in floor plans, which are used to intuitively display the design scheme of the exhibition hall. The division and classification of these three parts help in analyzing and evaluating the generation effect and practicability of the model in more detail.

## 2.3 Basic principles of the CGAN framework

The CGAN is an extension of the generative adversarial network (GAN) [62, 63], which is used for image generation tasks. The CGAN framework is similar to that of the GAN [64], consisting of a generator and a discriminator (Fig 4). The generator is responsible for generating images, and the discriminator is responsible for judging whether the generated images are real [65, 66]. The difference is that the CGAN needs to provide additional conditions, which can be any form of image (training A in Fig 4).

The proposed model architecture study is as follows. (1) Weight initialization: This function initializes the weights for the convolutional layer (Conv) and batch normalization layer (BatchNorm2d) in the network. A normal distribution is used to initialize the convolutional layer, a normal distribution is used for the batch normalization layer weights, and the bias term is set to 0.

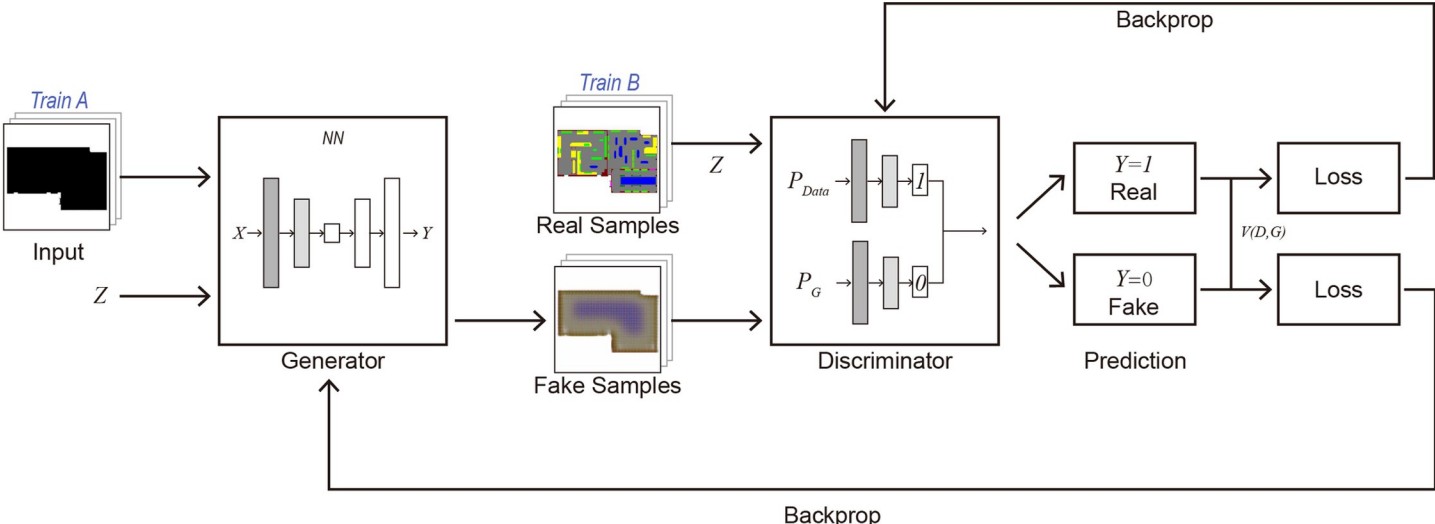

**Fig 4. Basic principles of the CGAN framework.**

(2) Normalization layer: This function selects different types of normalization layers, including batch normalization (batch) and instance normalization (instance). These normalization layers are used in convolutional neural networks to normalize layer inputs, helping to speed up the training process and reduce overfitting.

(3) Generator: The generator architecture provides a variety of configuration options, including a global generator, local enhancer and encoder. These options allow the generator structure to be tailored to specific application needs. The global generator uses multiple downsampling and upsampling layers and residual blocks to focus on generating the overall image. The local enhancer focuses more on local areas of the image to improve detail quality. The encoder may focus more on feature extraction. The generator uses normalization layers (batch or instance normalization) in different parts of the network and weight initialization via a weight initialization function, which allows the model to be trained from scratch without relying on pretrained models.

(4) Discriminator: The discriminator adopts a multiscale structure, which is called a multiscale discriminator. This design enables the discriminator to capture image features at different scales and effectively distinguish between real and synthetic images. The discriminator also contains multiple convolutional layers and uses normalization layers (batch or instance normalization) to improve training stability. The discriminator also performs weight initialization through the weight initialization function.

In terms of the loss function of the model, generator loss uses least squares GAN loss (LSGAN), feature matching loss (based on L1 loss), and VGG loss (perceptual loss based on the VGG network) (generator in Fig 4). The LSGAN loss uses the squared difference loss function for generator optimization, aiming to generate more realistic images while ensuring training stability. Feature matching loss and VGG loss are used to ensure that the generated image is similar to the real image at the feature and perceptual levels. Discriminator loss also uses LSGAN loss, which helps it effectively distinguish between real images and images generated by the generator (discriminator in Fig 4). This combination of multilevel loss functions helps generate high-quality images, ensuring authenticity and similarity at the pixel level and at more abstract features and perceptual levels. By using the LSGAN loss, the model can reduce

the vanishing gradient problem while improving the generated image quality, making it more suitable for high-resolution image generation tasks.

During the training process, the generator generates images according to the input conditions, and the discriminator judges whether the generated images are real (training B in Fig 4). The discriminator parameters are optimized through backpropagation, and a probability is output, indicating the probability that the input image is a real image [67]. The generator and the discriminator are trained alternately. By continuously optimizing the generator and discriminator parameters, the generator can generate an image as close as possible to a real image, and the discriminator can distinguish the real image from the generated image as well as possible [68]. After training, the generator can generate images with specific properties according to the input conditions. In this study, the model was trained from scratch without using pretrained models and parameters. Hyperparameters refer to the learning rate, batch size, number of epochs of training iterations, etc. These parameters affect the training process and performance of the model. The hyperparameters of this study were set to an initial learning rate of 0.001 (for detailed model training parameter settings, please refer to S4 File). In the 40 epochs after the 160th epoch, the learning rate began to drop to 0, so a total of 200 epochs were trained. In this paper, the CGAN was applied to generate an exhibition hall floor plan for an urban cultural museum, that is, an exhibition hall floor plan that meets certain conditions.

## 3. Results: Model training results and questionnaire survey results

### 3.1 Iterative process analysis of models and evaluation methods

In this study, the researchers used multiple indicators to evaluate the performance of the model, including:(1) Evaluation of the LOSS value during training: During the model's training, we recorded and examined the loss function value (LOSS) of the generative adversarial network's generator and discriminator to gauge the effectiveness of the training and the speed of the model's convergence. (2) Pixel similarity evaluation: After the model training is completed, we objectively assess the quality of the model-generated images based on the pixel similarity between the generated and real images. (3) In addition to the objective data evaluation mentioned above, we have also incorporated a subjective user evaluation. We invite users in professional fields to subjectively evaluate the generated museum floor plan through a user questionnaire. The evaluation content encompasses the image's visual effect, the rationality of the functional division, and the overall layout's aesthetics. Rating and feedback: We score the generated images based on user feedback and collect specific opinions and suggestions. By adopting multiple objective and subjective evaluation indicators, we strive to have a comprehensive understanding of the model's generation quality. These indicators include LOSS value during training, pixel similarity evaluation, and user evaluation. This multi-dimensional evaluation method can more comprehensively and accurately reflect the model's performance and generation effects.

**3.1.1 The loss log.** After the model is trained, the performance and learning of the model during training can be better understood by analyzing the loss log and observing the test pictures during the model iteration process. The loss log can reflect the change in the loss function value of the model during the training process [69]. If the value of the loss function continues to decrease, the model is gradually optimized during learning. If the loss function value does not decrease or fluctuate, the model has reached a learning bottleneck or has problems such as overfitting. In this study, the loss value (LOSS) of each generator and discriminator iteration during the model training process was determined to make a line graph (Fig 5). In the epoch cycle of Model 1 and Model 2, please refer to Appendices E and F for the detailed numerical statistics of G_GAN, G_GAN_Feat, G_VGG, D_real, and D_fake. Model 1 represents the

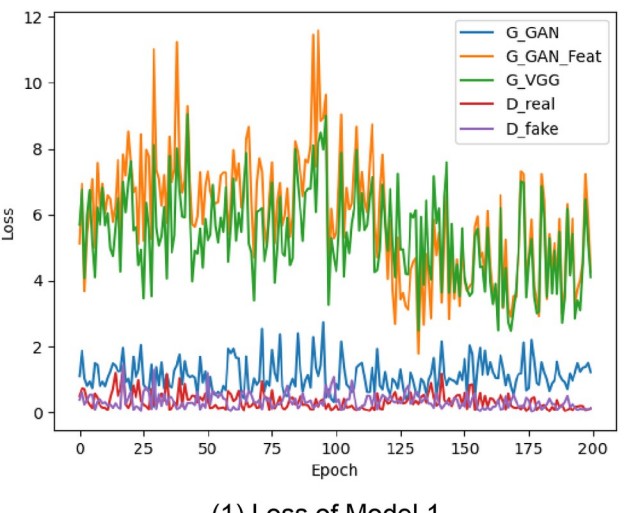 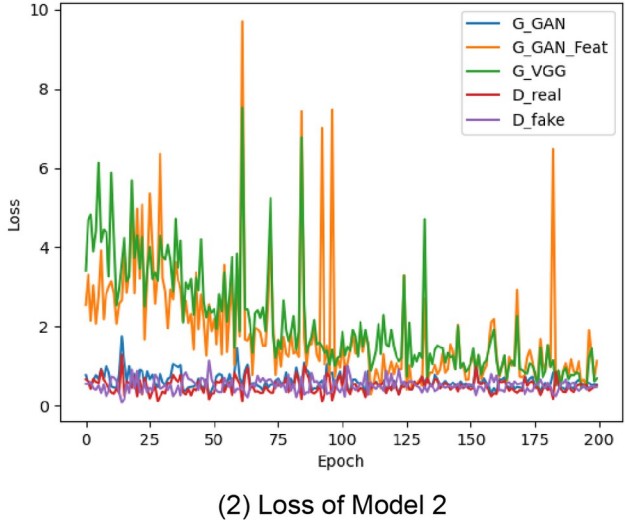

(1) Loss of Model 1

(2) Loss of Model 2

**Fig 5.** The learning curve for model training: (1) Loss of Model 1; (2) Loss of Model 2.

model from the FPP to the FSL, and Model 2 represents the model from the FSL to the FPE. The figure shows that after Model 1 and Model 2 were trained for 200 and 140 epochs, respectively, the loss value gradually decreased and stabilized in a lower value range, indicating that the performance and prediction accuracy of the models improved and gradually converged. The model loss log shows the following: (1) Model 1 requires more training iterations to converge, which indicates that the model has difficulty learning the relationship between the FPP and FSL. The reason is that the relationship between the FPP and FSL is complex or nonlinear, and the model may take longer to learn. (2) Model 2 requires a shorter iteration period, which shows that Model 2 has a better learning effect. The reason is that the relationship between the FSL and FPE is simple, and the model can learn it faster. (3) In comparison, a short iteration cycle usually indicates that the model has a better learning effect, but the specific accuracy and generalization ability of the model cannot be determined, and further verification is needed.

**3.1.2 Pixel similarity evaluation.** This study adopts a method based on image-pixel difference comparison to compare the differences between the generated CGAN images and the original images. The specific process and details of the method are as follows:

1. Image reading and color space conversion: OpenCV's imread function is used to read actual images and images generated by CGAN (real and generated images in Fig 6). Then, these two images are converted from the BGR color space to the RGB color space. This step is because OpenCV uses the BGR format by default, while usual image processing and display operations use the RGB format.

2. Difference image calculation: the pixel-by-pixel difference of two RGB images is calculated. This is done by calculating the difference between the corresponding pixels in the two images (using the cv2.absdiff function). The result is a difference image, where the value of each pixel represents how different the original two images are at that location.

3. Grayscale and binarization of difference images: To further analyze these differences, the script converts the difference images to grayscale images (using the cv2.cvtColor function) and then applies thresholding (using the cv2.threshold function). This thresholding step produces a binary image in which pixels that differ by more than a certain threshold are

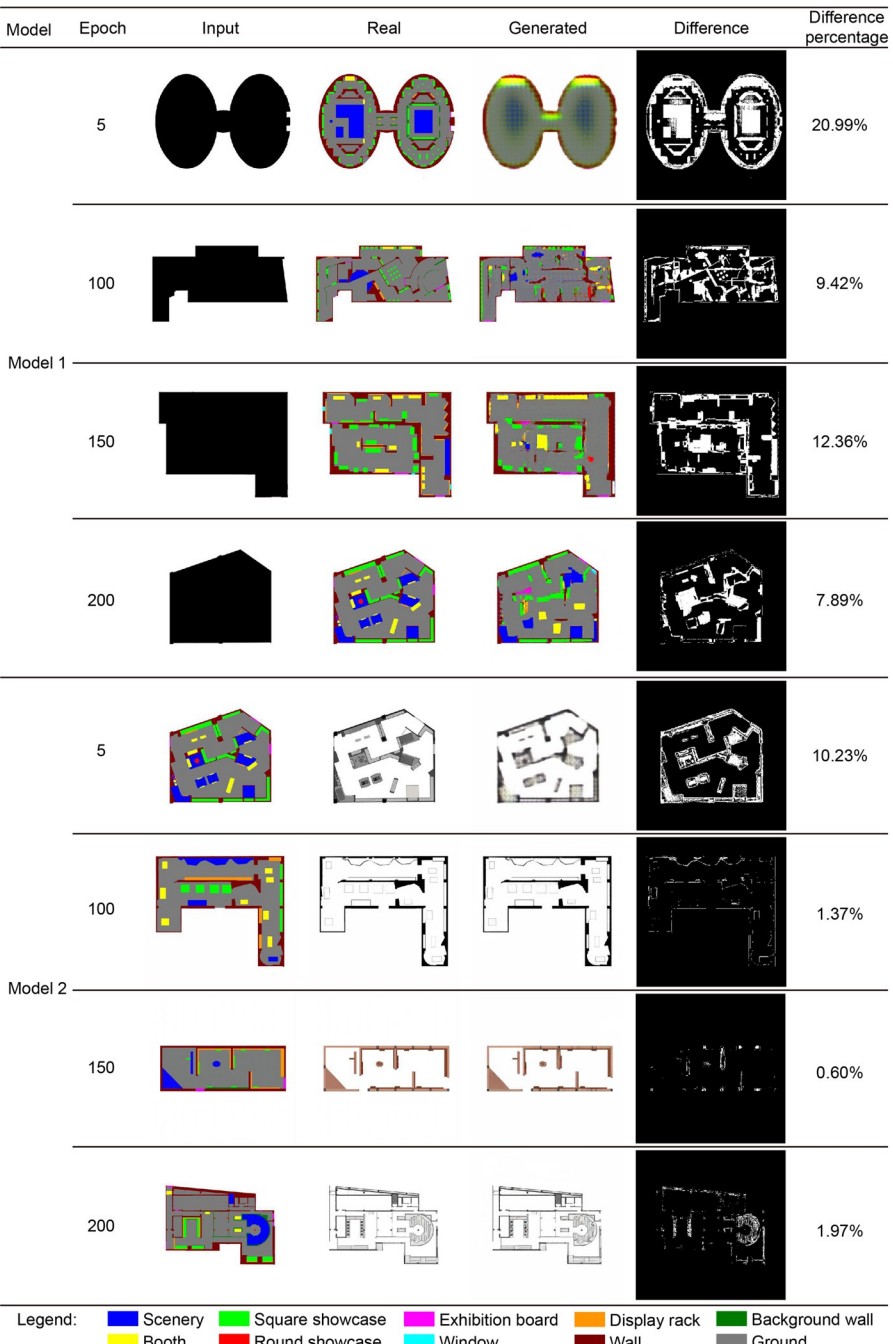

**Fig 6. Changes in model accuracy during epoch.**

marked white (indicating a significant difference) and the remaining pixels are black (the difference image in Fig 6).

4. Difference metric quantification: the number of white pixels in the binary difference image (using the np.sum function), which represents the number of pixels that are significantly different between the two images. Finally, this value is converted into a difference percentage (the difference percentage value in Fig 6).

By observing test images during the model iteration process using the above method, researchers can see whether the images generated by the model gradually become more realistic and clear. These questions will help researchers determine whether model training has converged. The model was tested at each iteration during model training, and test images were generated to determine changes in model accuracy (Fig 6). The test pictures of the model training process are as follows:

1. When Model 1 and Model 2 are trained for 200 epochs, the images generated by their generators are almost consistent with the real images (difference <10%), and the training goals are basically completed (accuracy >90%). This shows that Model 1 and Model 2 gradually learned the characteristics of the exhibition hall space design during the training process and successfully generated a real exhibition hall floor plan design.

2. However, the training process for Model 1 performs poorly. In the first 150 training epochs, there were many errors in the generated results, especially the layout of the display cabinets, which was quite different from the real images (the difference was 20.99% to 12.36%). This may be due to the high complexity of museum exhibition hall design and the weak regularity between various design solutions, which makes it difficult for Model 1 to capture effective design patterns in the early stages. This difference requires more training data and more complex model structures to resolve.

3. In comparison, the Model 2 training process is more stable. In the initial training stage, the test images show that the generator's effect has a smaller deviation from the real picture than Model 1 (the difference of the fifth-generation model is 10.23%). Subsequently, the model gradually improved its accuracy (discrepancy <2%) and steadily produced images that approximated real-world exhibition hall designs. This can be attributed to the fact that Model 2 changes from a FSL to a FPE to a lesser extent than Model 1 changes from a FPP to an FSL. The Model 2 tasks are relatively simple and regular, so the model can learn the characteristics of the exhibition hall design faster and generate more accurate design solutions.

In summary, Model 1 and Model 2 perform differently during training. Although Model 1 has some difficulties in the initial stage, it gradually improves the accuracy of the generated results after a long training period. Model 2 shows better stability and accuracy. Because the task is relatively simple, the model can quickly learn the rules and generate reasonable design solutions. To further improve model performance, more training data, more complex model structures, and the modeling and learning of more complex design rules may be needed.

### 3.2 CGAN model accuracy test and comparison

In this study, we didn't perform the traditional training set and test set division due to the limited amount of data (just 100 museum floor plans). Instead, we trained the model using the full amount of data, and evaluated its performance through cross-validation and manual pattern testing. This method ensures the model's reliability and generalization ability under limited data, effectively utilizes each sample's information, and achieves a comprehensive and reliable performance evaluation. After model training is complete, the model can be further evaluated by testing it with samples from the training set. This study employs CGAN to train two models with a sequential relationship: Model 1 is responsible for generating functional partitions from the outline of the museum floor plan; Model 2 generates the final museum floor plan based on the functional partitions generated by Model 1. This naming method can clearly highlight the sequential relationship between the two models, allowing readers to understand the model's

process and operation steps. In Fig 7, samples 1 to 3 are the test results of Model 1, and samples 4 to 6 are the test results of Model 2. The following conclusions can be drawn from this test:

1. The overall performance of Models 1 and 2 is good, and the generated pictures have no significant errors. This shows that the two models successfully learned the characteristics of the exhibition hall floor plan design during the training process and can generate a design that meets the actual requirements.

2. Model 1 has a slight distortion when generating exhibition hall furniture such as showcases and booths. This is because the regularity of the exhibition hall furniture layout is weak. In actual design projects, the arrangement of exhibition hall furniture is usually determined according to the size and quantity of exhibits rather than by the architect. Therefore, from the perspective of design drawings, furniture arrangement in the exhibition hall may lack obvious regularity. Nevertheless, Model 1 can still generate a relatively accurate graphic design of the exhibition hall. In future research, postprocessing techniques, such as image smoothing, edge enhancement, and denoising algorithms, will be used to reduce noise and distortion in generated pictures.

3. In comparison, Model 2 performs better. The task of Model 2 is to convert the pictures generated by Model 1 into easily recognizable floor plans for architects. The outputs of Model 2 can essentially serve as reference drawings for exhibition hall design after evaluation by researchers and qualified architects. This shows that Model 2 has certain accuracy and feasibility when transforming the generated design scheme into a practical and operable floor plan.

This study compares the effect of model generation with that of previous studies to highlight the main differences between this and previous studies [36]. Fig 8 shows selected plan profile pictures consistent with past studies used to test the model. These pictures are data that

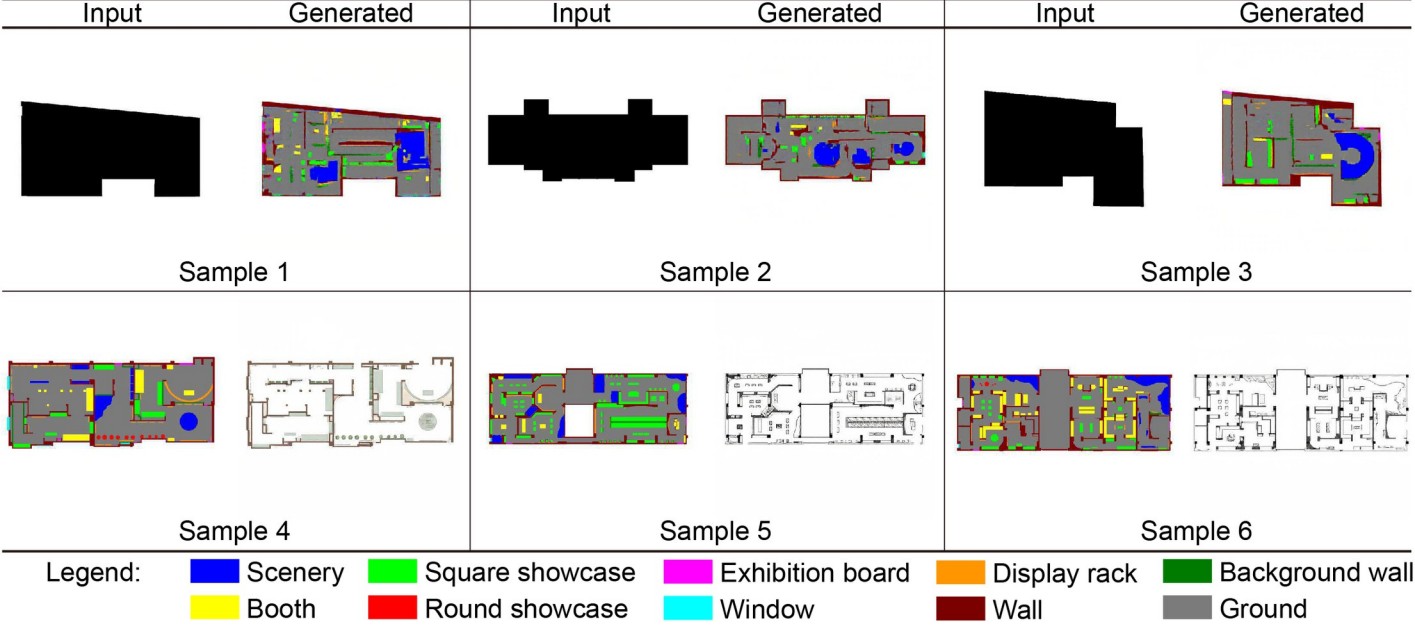

**Fig 7. Using the training samples to test the final model.**

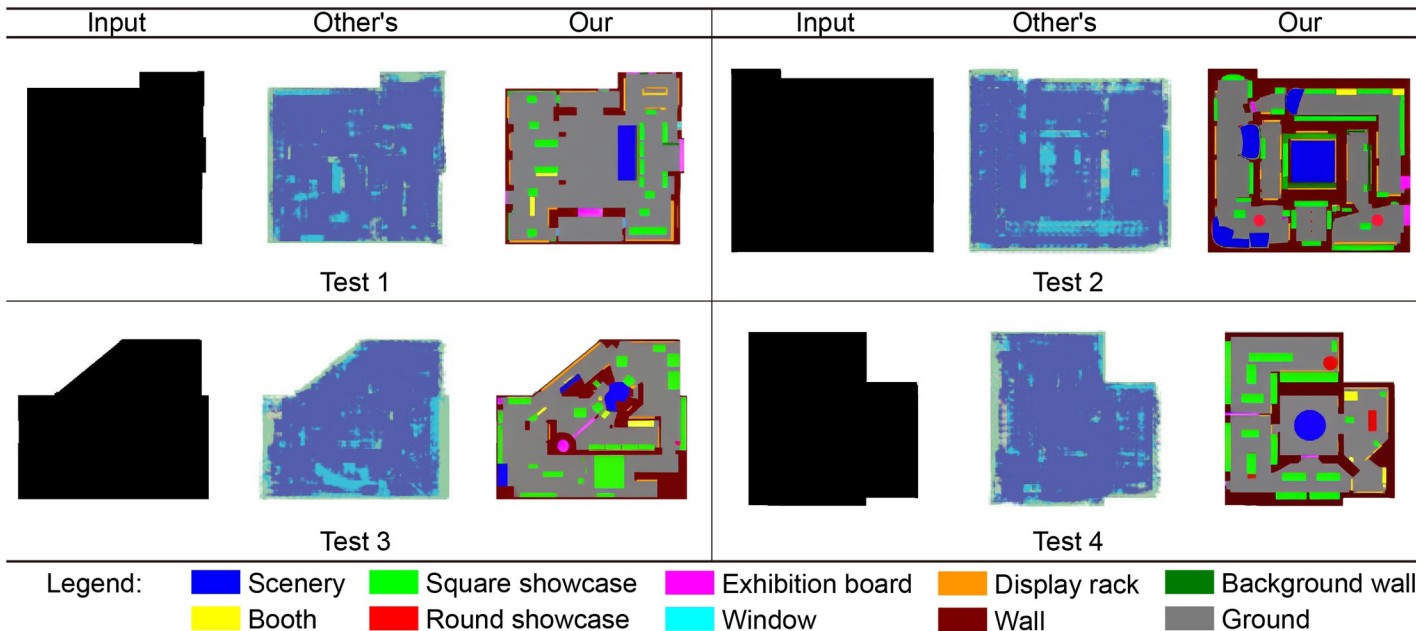

**Fig 8. Using the training samples to test the final model (The "others" in the figure refer to the models studied by other scholars).**

the model was not exposed to in the previous training process and were used to evaluate the performance of the most advanced model in this study.

Here are the test results: (1) Compared with previous studies, the proposed model shows significant improvement. This is because more training samples were added in this study (66 in previous studies, 100 in this study); the functions of the exhibition hall plan are marked in more detail (3 annotations in previous studies, 10 annotations in this study). The improvements in these respects made the model significantly better.

(2) In terms of the spatial division of the exhibition hall, the proposed Model 1 can generate clear spatial divisions, while the spatial divisions generated by the models in previous studies are relatively vague. This shows that the proposed model significantly improves the accuracy and clarity of space partitioning and can better meet the needs of space partitioning in exhibition hall design.

(3) In terms of furniture arrangement, the proposed Model 1 can generate more furniture, and there is also a significant improvement in quantity and type; the models in previous studies generate less furniture, and there are certain blank areas. This shows that the proposed model has a more detailed and richer furniture arrangement, can provide more diverse and complete furniture design schemes, and provides a larger creative space for exhibition hall design.

Model robustness testing, which aims to verify whether the model can generate multiple scenarios and test its performance on new materials, is an important part of model evaluation. This research selects the outline map of an exhibition hall shown in Fig 9 as the test object. With varying noise inputs to the model, Model 1 generated different design proposals, which were then transformed into intuitive planar renderings using Model 2. The figure shows that these generated schemes present significant differences in space division and furniture arrangement, providing greater creativity and flexibility for exhibition hall floor plan design and enabling architects to generate diverse designs according to different needs and scenarios.

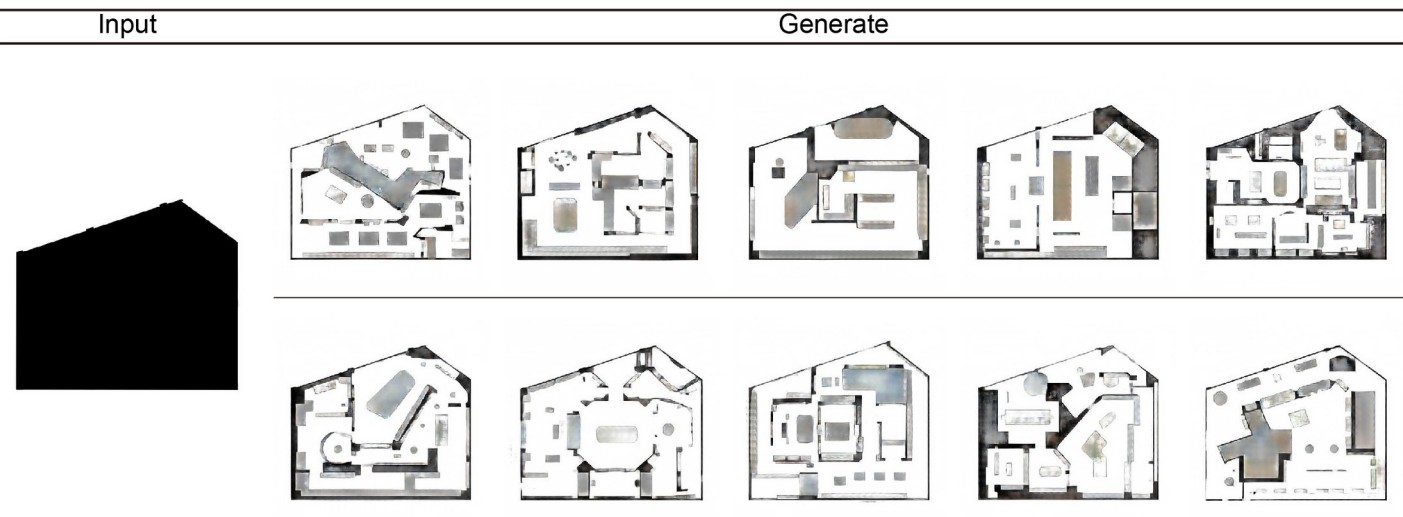

**Fig 9. The model is used to generate multiple design proposals for the outline of a museum exhibition hall.**

Model 1 generated different design proposals, which were then transformed into intuitive planar renderings using Model 2. Additionally, it provided personalized design solutions and creative inspiration.

In the new material testing part of the model, four planar profiles commonly found in museum exhibition halls are taken as the input conditions. Model 1 generates corresponding exhibition hall picture results based on these materials, while Model 2 uses the results generated by Model 1 as input conditions to generate a new design scheme (as shown in Fig 10).

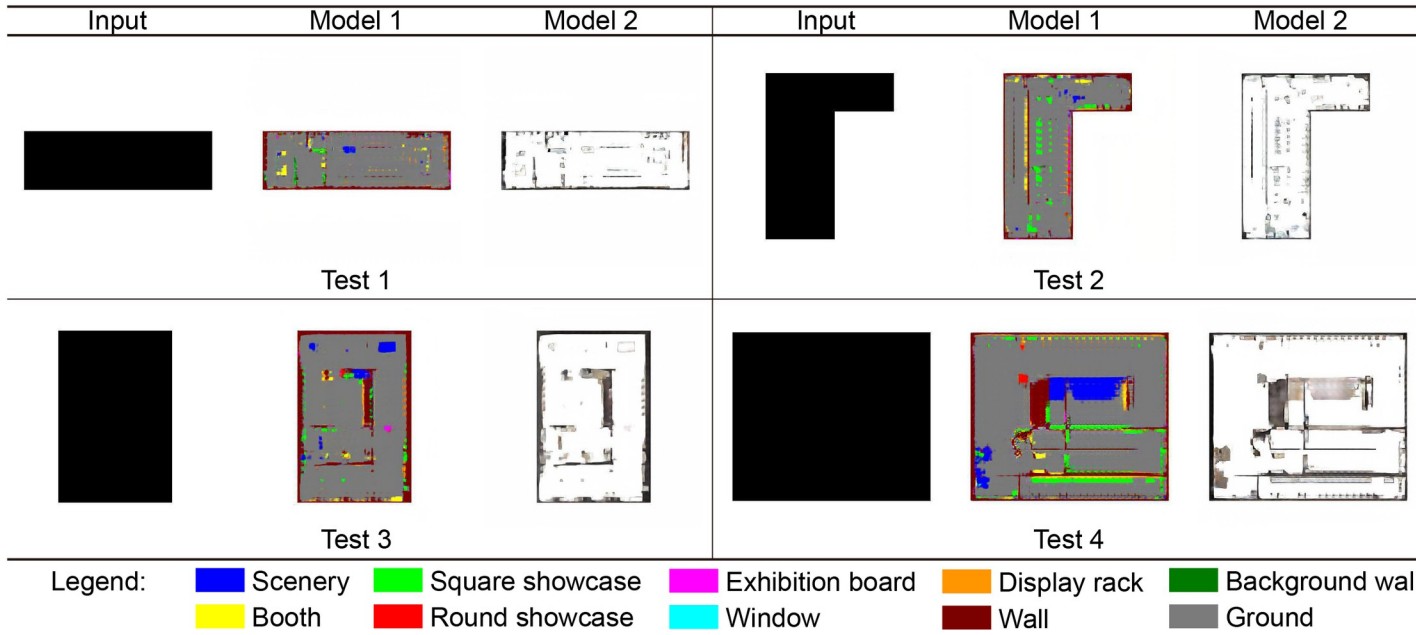

**Fig 10. Testing the generation effect of Models 1 and 2 using the new museum exhibition hall outline.**

From this experiment, the following observations can be drawn: (1) The overall generated results of Models 1 and 2 are better, and the generated elements have no obvious dislocation or distortion, showing high accuracy and reliability.

(2) In terms of wall generation, the model shows stable performance for different plane outline shapes of the exterior walls and partition walls of the exhibition hall, has strong space division capabilities, and can generate reasonable wall layouts according to the input conditions.

(3) In terms of interior furniture, the model can generate multiple types of furniture elements. In particular, for furniture with a large number and a large area, such as showcases and scenery, which are commonly used elements in the design of museum exhibition halls, the model can effectively generate furniture arrangements that meet the design requirements.

(4) However, there are still some problems with the proposed model. For example, in some cases, the model may divide the space into closed spaces or generate a small amount of furniture (test 1). These problems may be addressed by generating and selecting more reasonable solutions or by introducing corresponding adjustments during model training. Additionally, this suggests that the stability and reliability of the model may be further improved. In future research, the model needs to be further adjusted and optimized for this problem.

In summary, in testing model robustness, the proposed model performed well in generating exhibition hall design schemes; however, there are still some improvements to be made. These test results provide valuable feedback on the reliability and practicality of the model and suggest further challenges and directions for future research.

## 3.3 Analysis of public questionnaire scoring results

The researchers chose five groups of typical exhibition hall shapes, each group having two planes, to investigate the public's satisfaction with the CGAN-generated museum space layout. Among them, each group is based on the same shape, with one exhibition hall plan designed by an architect and the other one generated by the CGAN. Each question is worth 5 points, where 5 denotes strongly agree (very satisfied, very good) and 4 denotes relatively agree (relatively satisfied, relatively good). 3 represents general agreement (okay), 2 represents relative disapproval (not very good, not very satisfied), and 1 represents disapproval (not good, not satisfied). Without telling the respondents whether the floor plans were generated by the CGAN, the public was asked to rate these solutions according to their own preferences or professional abilities (see S7 File for the questionnaire).

**3.3.1 Questionnaire reliability and validity results.** A total of 297 questionnaires were distributed, and after removing 33 invalid questionnaires, 264 valid questionnaires were obtained. This study used SPSS 21 to analyze the reliability and validity of the questionnaire to verify the reliability of the questionnaire and the consistency of the data results. Reliability can reflect the stability of the questionnaire results and the true degree of the measured characteristics. Cronbach's alpha coefficient is currently the most common index used to evaluate the internal consistency of the questionnaires [70]. The value of the standardized Cronbach's alpha coefficient is between 0 and 1. The closer the coefficient is to 1, the higher the internal consistency of the questionnaire and the more reliable the results. It is generally believed that a questionnaire with a standardized Cronbach's alpha coefficient value greater than 0.6 is the most basic standard [71]. The reliability results of this questionnaire are shown in Table 1. The

Table 1. Reliability statistics of the questionnaire.

| Cronbach's Alpha | Cronbach's Alpha Based on Standardized Items | No. of Items |
|---|---|---|
| 0.821 | 0.821 | 10 |

Source: Author statistics

Cronbach's alpha coefficient of all measurement dimensions is 0.821, which is greater than the standard of 0.6, indicating that the overall reliability of the questionnaire is good.

This study analyzed the validity of the questionnaire for verification. Validity can describe the extent to which a scale accurately reflects the characteristics of the measurement object. The validity value (KMO) is between 0 and 1. The closer the value is to 1, the higher the authenticity of the measurement object [72]. This study mainly used the KMO and Bartlett sphericity tests to analyze the validity of the questionnaire and show whether the internal structure of the questionnaire was valid. Validity is generally considered to depend on the values of KMO and Bartlett. When the KMO test coefficient is >0.5, and the Bartlett value significance level is <0.05, the questionnaire has structural validity [73]. According to the results in Table 2, the KMO value of this questionnaire is 0.832, Bartlett's approximate chi-square is 729.605, the degree of freedom df is 45, and Bartlett's significance (Sig) is 0.000, less than 0.05, which meets the standard requirements. This shows that this questionnaire has good authenticity and validity.

**3.3.2 T-tests results.** According to the statistical results of all floor plans, without being informed of the source of the floor plans, the mean value of satisfaction for all floor plans is 3.261, which is greater than the 3-point (average) level. To further evaluate whether there is a significant difference between the architect's design and the CGAN model's design questionnaire results, the researchers used t tests and ANOVA hypothesis testing to verify the difference between the two.

For the architect's design results and the CGAN model's design, t tests are used. Before conducting t tests, the questionnaire data need to be classified and sorted. Among the 264 valid questionnaires, the data results of the architect's design and the CGAN model's design are classified and sorted. Since there are 5 groups (total 10) of shape comparisons in the questionnaire, the survey results of 264 people are divided into 2 groups (5 shapes in each group) to obtain the two groups of architect design (n = 1,360) and CGAN model design (n = 1,360) data. The results of testing it through SPSS 21 are as follows:

When P > 0.05, it indicates that there is homogeneity of variance between the two samples [74]. According to the results of the test of homogeneity of variance, based on the mean value P = 0.897 (P > 0.05), which meets the assumption of homogeneity of variances, further parameter testing can be performed (Table 3).

According to the t test results, the data show that the architect's design (M = 3.1848, SD = 0.95339) is significantly lower than the CGAN model's design (M = 3.3364, SD = 0.93319), t(98) = -4.126, p = 0.00, d = -0.15, 95% CI [-.22352; -.07951]. Among them,

Table 2. KMO and Bartlett's test for the questionnaire.

| Kaiser–Meyer–Olkin Measure of Sampling Adequacy | Bartlett's Test of Sphericity | | |
|---|---|---|---|
| | Approx. Chi-Square | df | Sig. |
| 0.832 | 729.605 | 45 | 0.000 |

Source: Author statistics

**Table 3. Test of homogeneity of variance.**

|  | Levene Statistic | df1 | df2 | P |
|---|---|---|---|---|
| Based on Mean | 0.017 | 1 | 2638 | 0.897 |
| Based on Median | 0.136 | 1 | 2638 | 0.712 |
| Based on Median and with adjusted df | 0.136 | 1 | 2625.815 | 0.712 |
| Based on trimmed mean | 0.028 | 1 | 2638 | 0.867 |

Source: Author statistics

p = 0.00 indicates that there is a significant difference between the architect's design and the CGAN model's design. t(98) = -4.126, indicating that the mean data of the CGAN model's design group is greater than the data of the architect's design group. It can be seen that the public has a higher overall recognition of the CGAN model's design (Tables 4 and 5).

**3.3.3 ANOVA (analysis of variance) test results.** For the comparison of differences between different shapes, an ANOVA test was used. The questionnaire data were classified and sorted by shape, and 5 sets of shape score data of the architect's design were obtained: square museum floor plan (n = 264), "T"-shaped museum floor plan (n = 264), oval museum floor plan (n = 264), rectangular museum floor plan (n = 264), and double-arc museum floor plan (n = 264). Five sets of shape score data for the CGAN model's design: square museum floor plan (n = 2 264), "T"-shaped museum floor plan (n = 264), oval museum floor plan (n = 264), rectangular museum floor plan (n = 264), and double-arc museum floor plan (n = 264).

Before the ANOVA test, it is necessary to test the homogeneity of variances in each group of samples. When the homogeneity of variance test result is P ≥ 0.05, there is a significant difference between the two groups. Multiple comparison methods are selected from the least significant difference (LSD) before proceeding to the ANOVA test [75]. The homogeneity of variance test result of the five groups of shape score data in the architect's design was P = 0.238 (P > 0.05), and the homogeneity of variance test result of the five groups of shape score data in the CGAN model's design was P = 0.051 (P > 0.05). This shows that there was a homogeneity of variances between the samples, which conforms to the assumption of homogeneity of variances, and further parameter testing was performed (Table 6).

ANOVA was used to test data from more than two groups. Generally, when P<0.05, there is a significant difference between each group [76]. In the five-group shape ANOVA test of the architect's design, the data show that the sum of squares between groups of the architect's design was 349.556 and the sum of squares within groups was 1,149.341, among which the sum of squares between groups was F = 14.175, P = 0.000 (P< 0.05), showing that there were significant differences between the shapes of the five groups of floor plans. The public could distinguish the floor plans of an architect's design and give different scores to each type of floor plan (Table 7).

In the multiple comparisons of the five groups of shapes in an architect's design, there were significant differences between the square museum floor plan and the other four planes, among which the difference between the square museum floor plan and the oval museum floor plan was the largest (P = 0.000). There was a significant difference between the two shapes of the "T"-shaped museum floor plan and square museum floor plan (P = 0.032) and the rectangular museum floor plan (P = 0.000). There was no significant difference between the "T"-shaped museum floor plan and the oval museum floor plan (p = 0.063) or the double-arc museum floor plan (p = 0.675). There were significant differences between the two shapes of the oval museum floor plan, the square museum floor plan (P = 0.000), and the rectangular

**Table 4. Group statistics.**

| | N | Mean | Std. Deviation | Std. Error Mean |
|---|---|---|---|---|
| Architect's design | 1320 | 3.1848 | 0.95339 | 0.02624 |
| CGAN model's design | 1320 | 3.3364 | 0.93319 | 0.02569 |

Source: Author statistics

museum floor plan (P = 0.000). There was no significant difference between the oval museum floor plan and the "T"-shaped museum floor plan (p = 0.063) or the double-arc museum floor plan (p = 0.149). There were significant differences between the rectangular museum floor plan and the other four floor plans. Compared with the square museum floor plan, P = 0.005, and the other three planes, P = 0.000. There was a significant difference between the double-arc museum floor plan and the square museum floor plan (P = 0.011) and the rectangular museum floor plan (P = 0.000). There was no significant difference between the double-arc museum floor plan and the oval museum floor plan (p = 0.149) or "T"-shaped museum floor plan (p = 0.675) (Table 8).

In summary, among the five floor plan shapes of the architect's design, the square museum floor plan has the most significant difference from other samples, followed by the rectangular museum floor plan. Although there is a significant relationship between the "T"-shaped museum floor plan, the oval museum floor plan, the double-arc museum floor plan, and other samples, the significance between them is relatively low, and there is still the possibility of improvement.

In the five-group plan shape ANOVA test of the CGAN model's design, the data show that the sum of squares between groups of the CGAN model's design is 34.988, and the sum of squares within the group is 1,113.667, among which the sum of squares between groups is F = 10.328, P = 0.000 (P<0.05). This shows that there are significant differences between the five groups of floor plan shapes. The public can distinguish the floor plans of the CGAN model's design and give different scores to each type of floor plan (Table 9).

In the multiple comparisons of five groups of floor plans with different shapes in the CGAN model's design, the square museum floor plan was compared with the "T"-shaped museum floor plan (P = 0.003), the oval museum floor plan (P = 0.000), and the double-arc museum floor plan. (P = 0.000) There are significant differences between these three shapes, but there is no significant difference from the rectangular museum floor plan (P = 0.108). There are significant differences between three shapes: "T"-shaped museum floor plans: square museum floor plans (P = 0.003), oval museum floor plans (P = 0.018), and oval museum floor

**Table 5. Independent samples test.**

| | Levene's Test for Equality of Variances | | t-test for Equality of Means | | | | | | | |
|---|---|---|---|---|---|---|---|---|---|---|
| | F | Sig. | t | df | Sig. (2-tailed) | Mean Difference | Std. Error Difference | 95% Confidence Interval of the Difference | | |
| | | | | | | | | Lower | Upper | |
| Equal variances assumed | 0.017 | 0.897 | -4.126 | 2638 | 0.000 | -0.15152 | 0.03672 | -0.22352 | -0.07951 | |
| Equal variances not assumed | | | -4.126 | 2636.792 | 0.000 | -0.15152 | 0.03672 | -0.22352 | -0.07951 | |

Source: Author statistics

**Table 6. Test of homogeneity of variances.**

|  | Architect's design | CGAN model's design |
|---|---|---|
| Levene Statistic | 1.382 | 2.365 |
| df1 | 4 | 4 |
| df2 | 1315 | 1315 |
| P | 0.238 | 0.051 |

Source: Author statistics

plans (P = 0.030). There is no significant difference between the "T"-shaped museum floor plan and the rectangular museum floor plan (p = 0.186). There are significant differences between the oval museum floor plan and square museum floor plan (P = 0.000), "T"-shaped museum floor plan (P = 0.018), and rectangular museum floor plan (P = 0.000). There is no significant difference between the oval museum floor plan and the double-arc museum floor plan (p = 0.850). There is a significant difference between the two shapes of rectangular museum floor plans, oval museum floor plans (P = 0.000), and double-arc museum floor plans (P = 0.000). There is no significant difference between a rectangular museum floor plan, a square museum floor plan (P = 0.108), and a "T"-shaped museum floor plan (P = 0.186). There are significant differences between the three shapes of the double-arc museum floor plan and square museum floor plan (P = 0.000), "T"-shaped museum floor plan (P = 0.030), and rectangular museum floor plan (P = 0.000). There is no significant difference between the double-arc museum floor plan and the oval museum floor plan (P = 0.850) (Table 10).

In summary, among the five floor plan designs of the CGAN model, the square museum floor plan, "T"-shaped museum floor plan, oval museum floor plan, and double-arc museum floor plan all have obvious differences from the other samples. The difference in significance between the rectangular museum floor plan and other samples is relatively low. This shows that the rectangular museum floor plan of the CGAN model's design needs to be improved and needs to be more distinctive than other types.

**3.3.4 Average result analysis of satisfaction.** Judging from the average statistical results of each question, most of the average satisfaction scores for the plans generated by the CGAN model are higher than the average scores for the plans designed by architects. Among the floor plans generated by the CGAN, the square museum floor plan has the highest average satisfaction score of 3.576, while the oval museum floor plan has the lowest score of 3.152. Among the floor plans designed by architects, the rectangular museum floor plan has the highest average satisfaction score of 3.511, and the oval museum floor plan has the lowest score of 2.955 (Table 11).

According to the comparative scores of different shape groups, the average scores of the four groups of shapes in the floor plan generated by the CGAN model are all higher than the scores of the floor plan designed by the architect. Only in the rectangular museum floor plan

**Table 7. ANOVA (Architect's design).**

|  | Sum of Squares | df | Mean Square | F | Sig. |
|---|---|---|---|---|---|
| Between Groups | 49.556 | 4 | 12.389 | 14.175 | 0.000 |
| Within Groups | 1149.341 | 1315 | .874 |  |  |
| Total | 1198.897 | 1319 |  |  |  |

Source: Author statistics

**Table 8. Multiple comparisons (Architect's design).**

| | | | Mean Difference (I-J) | Std. Error | Sig. | 95% Confidence Interval | |
|---|---|---|---|---|---|---|---|
| | | | | | | Lower Bound | Upper Bound |
| LSD | square museum floor plan | "T"-shaped museum floor plan | 0.1742* | 0.0814 | 0.032 | 0.015 | 0.334 |
| | | oval museum floor plan | 0.3258* | 0.0814 | 0.000 | 0.166 | 0.485 |
| | | rectangular museum floor plan | -0.2311* | 0.0814 | 0.005 | -0.391 | -0.071 |
| | | double-arc museum floor plan | 0.2083* | 0.0814 | 0.011 | 0.049 | 0.368 |
| | "T"-shaped museum floor plan | square museum floor plan | -0.1742* | 0.0814 | 0.032 | -0.334 | -0.015 |
| | | oval museum floor plan | 0.1515 | 0.0814 | 0.063 | -0.008 | 0.311 |
| | | rectangular museum floor plan | -0.4053* | 0.0814 | 0.000 | -0.565 | -0.246 |
| | | double-arc museum floor plan | 0.0341 | 0.0814 | 0.675 | -0.126 | 0.194 |
| | oval museum floor plan | square museum floor plan | -0.3258* | 0.0814 | 0.000 | -0.485 | -0.166 |
| | | "T"-shaped museum floor plan | -0.1515 | 0.0814 | 0.063 | -0.311 | 0.008 |
| | | rectangular museum floor plan | -0.5568* | 0.0814 | 0.000 | -0.716 | -0.397 |
| | | double-arc museum floor plan | -0.1174 | 0.0814 | 0.149 | -0.277 | 0.042 |
| | rectangular museum floor plan | square museum floor plan | 0.2311* | 0.0814 | 0.005 | 0.071 | 0.391 |
| | | "T"-shaped museum floor plan | 0.4053* | 0.0814 | 0.000 | 0.246 | 0.565 |
| | | oval museum floor plan | 0.5568* | 0.0814 | 0.000 | 0.397 | 0.716 |
| | | double-arc museum floor plan | 0.4394* | 0.0814 | 0.000 | 0.280 | 0.599 |
| | double-arc museum floor plan | square museum floor plan | -0.2083* | 0.0814 | 0.011 | -0.368 | -0.049 |
| | | "T"-shaped museum floor plan | -0.0341 | 0.0814 | 0.675 | -.194 | 0.126 |
| | | oval museum floor plan | 0.1174 | 0.0814 | 0.149 | -0.042 | 0.277 |
| | | rectangular museum floor plan | -0.4394* | 0.0814 | 0.000 | -0.599 | -0.280 |

*. The mean difference is significant at the 0.05 level.

Source: Author statistics

group is the average satisfaction score of the floor plan generated by the CGAN slightly lower than the average score of the architect's design (Fig 11): the plan designed by the architect had a score of 3.511, and that generated by the CGAN had a score of 3.447.

In summary, according to the survey statistics, the overall satisfaction level for the floor plans generated by the CGAN is better than that of the floor plans designed by architects. Among the floor plans generated by the CGAN, the square museum floor plan has the highest satisfaction level (3.576), and the oval museum floor plan has the lowest satisfaction level (3.152). Furthermore, the survey research results show that satisfaction with the rectangular museum floor plan designed by architects is higher than that generated by the CGAN model. This shows that in future research, in addition to the need to improve the oval museum floor plan, the generation of rectangular museum floor plans by the CGAN can be optimized.

**Table 9. ANOVA (CGAN model's design).**

| | Sum of Squares | df | Mean Square | F | P |
|---|---|---|---|---|---|
| Between Groups | 34.988 | 4 | 8.747 | 10.328 | 0.000 |
| Within Groups | 1113.667 | 1315 | 0.847 | | |
| Total | 1148.655 | 1319 | | | |

Source: Author statistics

**Table 10. Multiple comparisons (CGAN model's design).**

| | | | Mean Difference (I-J) | Std. Error | P | 95% Confidence Interval | |
|---|---|---|---|---|---|---|---|
| | | | | | | Lower Bound | Upper Bound |
| LSD | square museum floor plan | "T"-shaped museum floor plan | 0.2348* | 0.0801 | 0.003 | 0.078 | 0.392 |
| | | oval museum floor plan | 0.4242* | 0.0801 | 0.000 | 0.267 | 0.581 |
| | | rectangular museum floor plan | 0.1288 | 0.0801 | 0.108 | -0.028 | 0.286 |
| | | double-arc museum floor plan | 0.4091* | 0.0801 | 0.000 | 0.252 | 0.566 |
| | "T"-shaped museum floor plan | square museum floor plan | -0.2348* | 0.0801 | 0.003 | -0.392 | -0.078 |
| | | oval museum floor plan | 0.1894* | 0.0801 | 0.018 | 0.032 | 0.347 |
| | | rectangular museum floor plan | -0.1061 | 0.0801 | 0.186 | -0.263 | 0.051 |
| | | double-arc museum floor plan | 0.1742* | 0.0801 | 0.030 | 0.017 | 0.331 |
| | oval museum floor plan | square museum floor plan | -0.4242* | 0.0801 | 0.000 | -0.581 | -0.267 |
| | | "T"-shaped museum floor plan | -0.1894* | 0.0801 | 0.018 | -0.347 | -0.032 |
| | | rectangular museum floor plan | -0.2955* | 0.0801 | 0.000 | -0.453 | -0.138 |
| | | double-arc museum floor plan | -0.0152 | 0.0801 | 0.850 | -0.172 | 0.142 |
| | rectangular museum floor plan | square museum floor plan | -0.1288 | 0.0801 | 0.108 | -0.286 | 0.028 |
| | | "T"-shaped museum floor plan | 0.1061 | 0.0801 | 0.186 | -0.051 | 0.263 |
| | | oval museum floor plan | 0.2955* | 0.0801 | 0.000 | 0.138 | 0.453 |
| | | double-arc museum floor plan | 0.2803* | 0.0801 | 0.000 | 0.123 | 0.437 |
| | double-arc museum floor plan | square museum floor plan | -0.4091* | 0.0801 | 0.000 | -0.566 | -0.252 |
| | | "T"-shaped museum floor plan | -0.1742* | 0.0801 | 0.030 | -0.331 | -0.017 |
| | | oval museum floor plan | 0.0152 | 0.0801 | 0.850 | -0.142 | 0.172 |
| | | rectangular museum floor plan | -0.2803* | 0.0801 | 0.000 | -0.437 | -0.123 |

*. The mean difference is significant at the 0.05 level.

Source: Author statistics

# 4. Discussion: Contrasting the model's output with the architect's designs

## 4.1 Comparison of output scheme designs

A comparative analysis was conducted with an architect's floor plan design for the exhibition hall to further assess and confirm the effect of the model. This can determine the practicability of the model and show whether the model can provide new design directions, layout ideas, and creative inspiration, thereby enriching the architect's program choices and providing a new perspective. First, in terms of experimental materials, this study collected five floor plans

**Table 11. Item statistics.**

| Group No. | Shape Group | Source | Mean | Std. Deviation | No. of Items |
|---|---|---|---|---|---|
| 1 | Square Museum Floor Plan | Architect's design | 3.280 | 0.9377 | 264 |
| | | CGAN model's design | 3.576 | 0.8988 | 264 |
| 2 | "T" Shaped Museum Floor Plan | Architect's design | 3.106 | 0.9332 | 264 |
| | | CGAN model's design | 3.341 | 0.9419 | 264 |
| 3 | Oval Museum Floor Plan | Architect's design | 2.955 | 0.9259 | 264 |
| | | CGAN model's design | 3.152 | 0.8675 | 264 |
| 4 | Rectangular Museum Floor Plan | Architect's design | 3.511 | 0.9512 | 264 |
| | | CGAN model's design | 3.447 | 0.9458 | 264 |
| 5 | Double Arc Museum Floor Plan | Architect's design | 3.072 | 0.9262 | 264 |
| | | CGAN model's design | 3.167 | 0.9446 | 264 |

Source: Author statistics

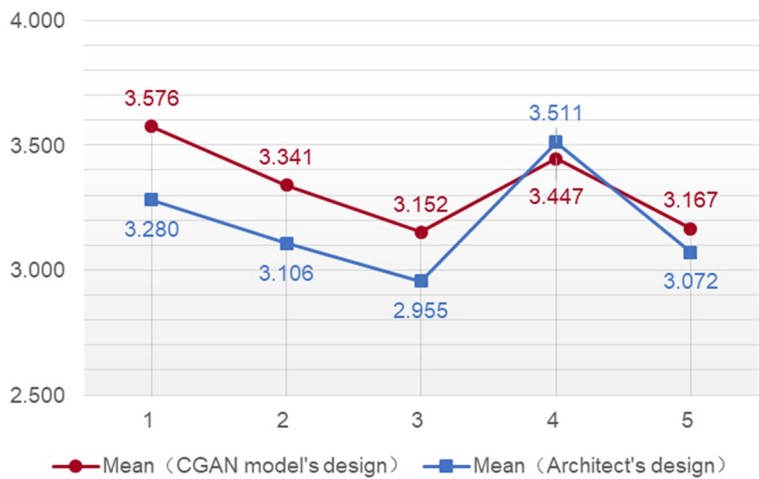

**Fig 11. Average scores of satisfaction for the 5 groups of floor plan.**

of exhibition halls designed by an architect. The floor plans of these exhibition halls have a relatively common form among Chinese museum exhibition halls. Polygons, partially raised polygons, circles, polygons with external stairs, and double-arc shapes (Fig 12, projects 1 to 5) are included to ensure material diversity and representation. The experimental process and results are shown in Fig 12. The plan designed by the architect is processed into a plan outline and used as the input material for Model 1. The generated image of Model 1 is further used as the input material for Model 2 to generate the final plan. In addition, to more comprehensively display the design effect and objective evaluation and comparison, in addition to the questionnaire survey mentioned above (refer to Section 3.3 Analysis of Public Questionnaire Scoring Results), the researchers also referred to the model designs and drawings designed by these architects and performed modeling processing (Fig 13).

This study found the following: (1) The effect of model generation is compared with the architect's design. The model can use experimental materials to generate and design the floor plan of the exhibition hall and finally present a clear space division and reasonable furniture arrangement. For example, in Project 4, the floor plan of the exhibition hall is connected with a circular staircase, and this feature is successfully identified in the model. However, it also has common effects of AI images, such as slight blurring and distortion in some details, which can clearly distinguish the model from the architect's floor plan.

(2) The CGAN model's design and the architect's design have significant differences. For example, in Project 3 and Project 5, the space divisions and furniture layouts of the exhibition halls in the two designs are quite different. Architects are better at making use of the characteristics of planar outlines to improve the utilization of space and avoid wasting corner space. However, the CGAN model responds to various planar forms with a vertical or horizontal space division, which may lead to a waste of space.

(3) The two design schemes also have some similarities. In some conventional exhibition hall plan forms, such as Project 1, Project 2, and Project 4, the effects of model design and artificial design are approximate. This shows that the model has better design ability for this kind of simple, noncurved polygonal plane. However, because the model is better at vertical or horizontal space division, it is not as flexible as the architect's design. For example, in

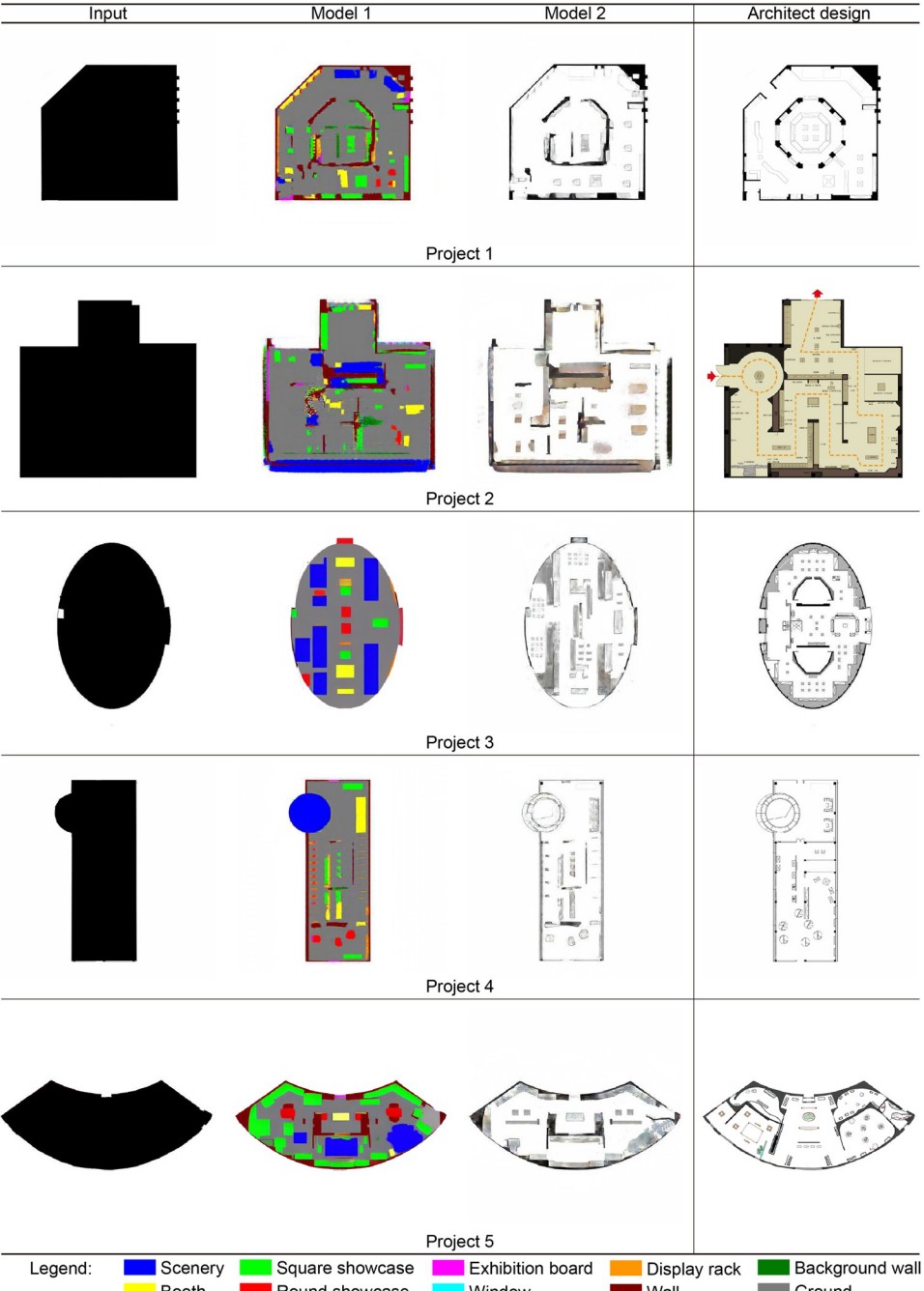

**Fig 12. Result comparison between the architect design and CGAN model generative design.**

Project 2, the architect designed a circular space that was not represented in the model design.

(4) The main problems with the model design scheme include errors in the layout of entrances and exits in the plan, weak adaptability to the planar outline of the surface, and low clarity and cleanliness of the generated image effect. These problems necessitate further improvement and optimization.

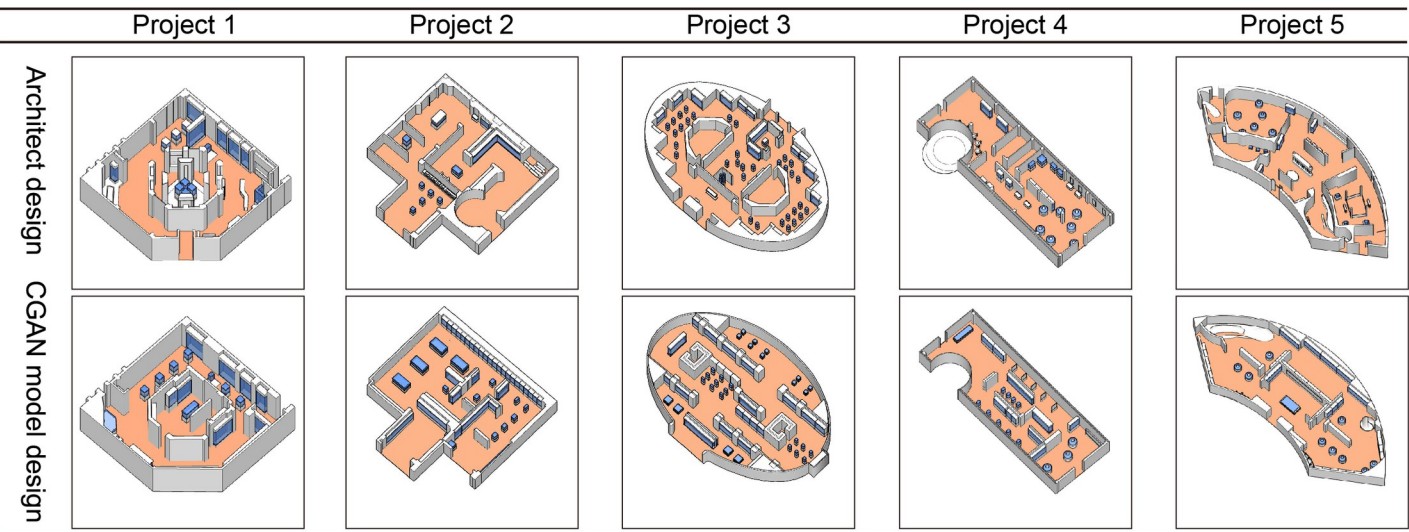

**Fig 13. 3D model comparison of the architect design and CGAN model generative design.**

(5) In modeling, the researchers make reference to the architect's plan and the CGAN model. In the 3D model, the schemes designed by the architect and the CGAN model have different characteristics. The CGAN model design scheme can also be adjusted during practical application, and some small errors may be corrected during the modeling process by the researchers. Therefore, the quality of the modeled scheme and the CGAN model design scheme can be very close to the architect's design level.

## 4.2 Multidimensional application and impact of machine learning in museum exhibition hall design

In the design process for museum exhibition halls, the presentation of curatorial content and the shaping of audience experience are often regarded as core elements [77]. In this context, it is also crucial to explore and reveal how machine learning can balance and fully reflect the importance of these two elements when assisting in the generation of design solutions. In the process of generating design proposals, the model adopted in this study not only examines the functionality and aesthetic expression of the space but also emphasizes the consistency of the design proposals with the curatorial themes and goals and its contribution to enriching the audience experience. For example, when the model generates design plans, it uses algorithms to generate many plans for the streamlined layout of exhibits. The designer needs to consider that the audience should enjoy a smooth and natural visiting experience, as well as full participation and interaction while visiting each exhibition area, and select the most suitable solution. Through human-computer interaction between this artificial intelligence model and the designer, the lack of emotional and subjective factors in the model design process can be compensated for.

When machine learning methods are placed in the context of global multiculturalism, their unique value and potential for different cultural backgrounds can be discovered. The design principles and aesthetic expression of museum exhibitions are often deeply influenced and shaped by specific cultural backgrounds [78]. In this study, the machine learning model showed the potential to generate design solutions that fit this cultural context by analyzing and learning from 63 design cases for the same type of urban cultural museum in China.

Furthermore, machine learning-assisted design methods can have an impact on design practice. This new approach not only provides designers with an efficient tool to help them generate and explore multiple design options in less time but also leaves them more energy to focus on other key design aspects, such as the accurate presentation of curatorial content, the in-depth exploration of storytelling, and the meticulous refinement of the audience experience. In addition, the application of machine learning brings new perspectives and possibilities for design practice, injecting new vitality into the diversity and innovation of museum exhibitions. In future research and practice, the application of machine learning in museum exhibition hall design can be further broadened, not only contributing to the aesthetics and functionality of design but also playing a unique role in the blending and dialog of global multiculturalism.

## 4.3 The shortcomings of the current floor plan generation

The research for this project only involves the spatial layout design issues of museum display design, which play a certain supporting role in exhibition design. Although this article has made some practical explorations, the systematicness and relevance of the theory still need to be deepened, and the technical aspects have not been accurately grasped. More mature cases and practical experience are needed to verify. Although artificial intelligence is in full swing in museums, as scholar Su Donghai said, digital technology relies more on its vivid display characteristics to assist and supplement traditional display methods [79]. Digital technology cannot surpass it, let alone completely replace it.

1. It is impossible to understand the text content of the exhibition outline because the artificial intelligence-powered spatial layout of the museum does not correspond to the content of the exhibition outline. It is limited to the design of the exhibition and streamlines the museum's spatial layout through machine learning. The museum's spatial layout and exhibition circulation design must be designed in conjunction with the actual content of the exhibition schedule. Therefore, it is also hoped that the next step of research can allow the deep neural network system to imitate the human brain and interpret cultural relics and text content through deep learning and probabilistic modeling. Let artificial intelligence build a knowledge graph, and then classify and optimize the content in the exhibition outline. At the same time, combining narrative methods with artificial intelligence is a new challenge for museums in the future. Artificial intelligence participates in museum exhibition narrative methods that are different from the previous straightforward and space-dominated design models. Instead, it relies on artificial intelligence's data analysis and algorithmic operations to create hierarchical exhibition content to form a spatial layout that can stimulate the audience's interest and emotional resonance. This also encourages the museum to provide visitors with an immersive and more interactive visiting experience and expand the boundaries of exhibitions to attract more viewers' attention.

2. This design research result is suitable for assistant-level display designers because designers at this stage do not have much experience in controlling the floor plan layout design of a museum. It is also suitable for the design of some relatively urgent museum projects. It can provide some floor layout drawings for designers to filter, find the plane that matches the text, and then optimize it.

3. The design research results do not form parameters that can be set to facilitate the setting of the museum's layout. For example, if the preface hall in a museum is to be set up in a specific location, the comfort of the audience during viewing should be taken into consideration, and then a matching museum floor plan should be generated.

4. This design research result cannot restrict the flow of people in the museum at this stage. For example, the width of the pedestrian flow route should be no less than 2.5 meters, forming a single-channel pedestrian flow route, a multi-channel pedestrian flow route, or a pedestrian flow route extending in all directions.

5. The floor plan generated as a result of this design research is in the preliminary stage of early conceptual design and cannot take into account the details of the museum's floor plan, such as the corner closing issues in the floor plan layout. It also failed to meet the problem of matching booths and display panels in museum space design, and it took time to optimize the generated floor plan in the later stage.

6. The design of the space layout of the artificial intelligence-powered museum does not take into account the contextual experience of the audience. In the process of creating public cultural service situations in museums, the knowledge structure and identity background of the audience should be considered, and more considerate services should be provided to users through adaptive experience situations. From the beginning of the design of the museum ranking layout, the interaction between visitors and exhibits was taken into consideration, forming a two-way interaction between the museum content and the audience, prompting the audience to have ups and downs of emotional reactions during the viewing process.

## 5. Conclusion

This research explores the method of automatically generating floor plans for museum exhibition halls to provide architects with auxiliary tools and creative inspiration in the design process. In the process of training, testing, and evaluating the model, the paper describes and validates the research methodology in detail. The main conclusions of the study are as follows:

1. An automatic generation model of floor plans of museum exhibition halls was successfully developed through the CGAN method. The model can generate diverse and reasonable exhibition hall schemes, providing architects with practical results and design inspiration. The extensive experiments and evaluation results show that the model can generate proposals with clear space division and reasonable furniture layout and can accurately identify and process the features and details of the exhibition hall.

2. A comparison of the CGAN model's designs with an architect's artificially created exhibition hall floor plans reveals differences between the two. The CGAN model is more suitable for simple nonsurface polygonal planes, but it needs to be improved for complex surface shapes and detail processing. At the same time, through public questionnaire statistics, it was found that the overall satisfaction level of the floor plan generated by the CGAN is better than that designed by the architect. The square museum floor plan has the highest satisfaction level (3.576), and the oval museum floor plan has the lowest satisfaction level (3.152). The satisfaction level of the architect's rectangular museum floor plan is higher than that of the CGAN model's generated floor plan, demonstrating the need to enhance and optimize the CGAN model's training for the oval museum floor plan.

3. Although the model has some limitations and deficiencies, such as the need for improvement in image clarity and detail processing capabilities, through comparison and analysis, the research shows that the model has potential and value in assisting architects in the design process. It also has the potential to be applied in similar fields, such as office space design, home decoration space design, landscape space design, historical arcade design, and building renovation design.

This study has the following shortcomings: (1) The detail processing ability of the model needs to be improved so that it can generate clear and realistic image results more accurately. (2) There are still some challenges for the model when dealing with the plane profile of the surface, which needs further improvement and optimization. (3) The generated image effect still has room for improvement in terms of clarity and cleanliness. (4) This research is mainly aimed at exhibition halls for urban culture, and its universality in nonurban cultural museums needs to be further verified, such as by considering a small-scale exhibition hall renovated from an old residential house in the countryside.

Finally, future research directions include (1) further improving and optimizing the CGAN model to increase its robustness and stability to meet more complex and diverse exhibition hall design needs; and (2) introducing more professional data samples and annotation information from the field of architecture to enrich the design diversity and creative generation capabilities of the model. (3) In order to enable the model to adapt to changing design trends and user needs, future research can use incremental learning and transfer learning so that the model can gradually learn new data and new features without the need for complete retraining. For example, incremental learning enables the model to gradually accumulate knowledge, while transfer learning uses pre-trained models to quickly adapt to new design trends and needs through fine-tuning. Both methods can effectively reduce the computational cost of each training session. (4) exploring collaborative design methods with models and architects to improve design efficiency and quality and achieve better human-computer interaction; and (5) conducting in-depth studies and comparisons of museums not related to urban culture, such as science and technology museums, which focus on experience-based installations and equipment. Importantly, in the era of digital humanities, it is necessary not only to digitize ancient and precious research materials but also to explore digitization and intelligence in museum exhibition design and space layout.

## Supporting information

**S1 File. Machine learning environment configuration.**
(DOCX)

**S2 File. Research materials (museum floor plan) source statistics.**
(DOCX)

**S3 File. Detailed RGB values in label making.**
(DOCX)

**S4 File. Detailed parameter settings for model training.**
(DOCX)

**S5 File. Numerical statistics during the Model 1 epoch period.**
(DOCX)

**S6 File. Numerical statistics during the Model 2 epoch period.**
(DOCX)

**S7 File. Museum space layout design satisfaction questionnaire.**
(DOCX)

## Acknowledgments

The author would like to thank Guangdong Jimei Design Engineering Company, Hangzhou Zhengye Decoration Design Co., Ltd., Shenzhen Silk Road Cultural Development Co., Ltd.,

and Guangzhou Litian Exhibition Design Engineering Co., Ltd. for providing data collection assistance during the research process. At the same time, the author would also like to thank five students, Chen Yekun, Wu Peijian, Su Houbo, Zhang Yueting, and Song Wentao, for their assistance during the field survey and questionnaire distribution.

## Author Contributions

**Conceptualization:** Yile Chen.

**Data curation:** Qiang Tang.

**Formal analysis:** Liang Zheng, Yile Chen.

**Funding acquisition:** Qiang Tang.

**Investigation:** Qiang Tang.

**Methodology:** Junzhang Chen.

**Resources:** Yile Chen.

**Software:** Liang Zheng.

**Validation:** Lina Yan.

**Visualization:** Yile Chen, Lina Yan.

**Writing – original draft:** Liang Zheng, Yile Chen.

**Writing – review & editing:** Liang Zheng, Yile Chen.

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
