## [Decision Letter · Decision Letter 0]

22 Jul 2024

PONE-D-24-12619Artificial Intelligence Empowering Museum Space Layout Design: Insights from ChinaPLOS ONE

Dear Dr. Chen,

Thank you for submitting your manuscript to PLOS ONE. After careful consideration, we feel that it has merit but does not fully meet PLOS ONE’s publication criteria as it currently stands. Therefore, we invite you to submit a revised version of the manuscript that addresses the points raised during the review process.

We look forward to receiving your revised manuscript.

Kind regards,

Sushank Chaudhary, Ph.D

Academic Editor

PLOS ONE

“This study was supported by the research from the Guangdong Provincial Department of Education’s key scientific research platforms and projects for general universities in 2023: Guangdong, Hong Kong, and Macao Cultural Heritage Protection and Innovation Design Team (Funding Project Number: 2023WCXTD042; Shunde Polytechnic Internal Number: 2023-KJZX047); 2020 Guangdong Province Ordinary Universities Characteristic Innovation Project (No. 2020WTSCX286); 2019 Guangdong Province Ordinary University Philosophy and Social Science Project (No. 2019GXJK131); Shunde Polytechnic "Tang Qiang Cantonese Area Cultural Heritage Protection Design Skills Master Studio.”

Reviewers' comments:

Reviewer's Responses to Questions

**Comments to the Author**

1. Is the manuscript technically sound, and do the data support the conclusions?

Reviewer #1: Partly

Reviewer #2: Yes

2. Has the statistical analysis been performed appropriately and rigorously? 

Reviewer #1: Yes

Reviewer #2: Yes

3. Have the authors made all data underlying the findings in their manuscript fully available?

Reviewer #1: No

Reviewer #2: Yes

4. Is the manuscript presented in an intelligible fashion and written in standard English?

Reviewer #1: Yes

Reviewer #2: Yes

5. Review Comments to the Author

Reviewer #1: This paper mainly studies the use of generative adversarial networks (GAN) to design the spatial layout of museums. The author's ideas are very creative. Because artificial intelligence is widely used in various fields. From the perspective of methodology, this paper only brings the very mature model in the field of artificial intelligence to this problem, but does not make any improvement in combination with the problem. In my opinion, this paper needs further improvement. Specific suggestions are as follows:

(1) Why does the author choose conditional generative adversarial network (CGAN) instead of other AI models? In terms of method, CGAN is not the latest generative model. What advantages does it have in solving this problem compared with other generative models?

(2) As can be seen from the text, the author only selected 100 data for training. This amount of data is relatively small in the field of artificial intelligence. The authors also mention that no pre-trained model was used. So how do you make sure that this amount of data is sufficient for the model to be trained?

(3) In what proportion is the data divided into the training and test sets?

(4) Chapter Settings and text descriptions are somewhat confusing. The article mentions "Model 1" and "Model 2" several times. Why not give them specific names to distinguish them better?

(5) section 3.2 and 3.3 can be combined into one section.

(6) The metrics used by the authors are not commonly used in artificial intelligence. It is suggested to add artificial intelligence evaluation index of image generation performance.

(7) Who is the "other" in Figure 8? Please indicate it clearly in the drawing.

Reviewer #2: In this work, author presented an innovative approach to using artificial intelligence, specifically the CGAN model, to enhance the space design of exhibition halls in urban cultural museums. The study addresses the limitations of traditional space design methods and highlights the potential benefits and application prospects of integrating AI into the design process. Here is the point author may want to highlight:

The static nature of trained models may limit their adaptability to evolving design trends and changing user needs. Continuous updating and retraining of the model with new data are essential to keep the generated designs relevant and up-to-date. Is it possible to make adaptable according to changed requirement? Every time training and testing with new data will increase the computational and cost both.

6. PLOS authors have the option to publish the peer review history of their article (what does this mean?). If published, this will include your full peer review and any attached files.

Reviewer #1: No

Reviewer #2: No

---

## [Author Response · Author response to Decision Letter 0]

2 Aug 2024

Thank you very much for your recognition of our research and your comments. We also noticed that the format requirements and financial disclosure requirements have been revised. The format requirements have been adjusted, including the font size and image citation statement. The financial disclosure has been added in the cover letter. The following is our revised response:

Reviewer #1: This paper mainly studies the use of generative adversarial networks (GAN) to design the spatial layout of museums. The author's ideas are very creative. Because artificial intelligence is widely used in various fields. From the perspective of methodology, this paper only brings the very mature model in the field of artificial intelligence to this problem, but does not make any improvement in combination with the problem. In my opinion, this paper needs further improvement. Specific suggestions are as follows:

(1) Why does the author choose conditional generative adversarial network (CGAN) instead of other AI models? In terms of method, CGAN is not the latest generative model. What advantages does it have in solving this problem compared with other generative models?

Response: Thank you for your comments. 

The conditional generative adversarial network (CGAN) is an extension of the generative adversarial network (GAN). It adds conditional variables to the input, ensuring that the generated samples not only conform to the data distribution but also meet specific conditional requirements. Compared to GAN and stable extension models, CGAN can better control the generation results, resulting in samples that meet specific requirements. Specifically, it has the following advantages:

(a) Ensuring precise control over the generation process is crucial. CGAN permits the imposition of conditional constraints during the generation process. These conditions, including the outline and functional zoning of the museum floor plan, are included in this study. This conditional control ensures that the generated floor plan not only conforms to the museum's structural outline but also meets the functional zoning requirements, resulting in accurate floor plan generation.

(b) Better handling of complex structures. Museum floor plans usually contain complex structures and diverse functional zoning. CGAN performs well in handling such complex structures because it can learn the complex patterns of data distribution through the adversarial training process and generate high-quality samples. In contrast, other generative models, such as variational autoencoders (VAE), may face greater challenges in handling complex structures.

(c) Existing successful cases and experience. CGAN has many successful applications in image generation and image restoration. For example, in image restoration tasks, CGAN can generate high-quality image details significantly better than traditional methods. Therefore, choosing CGAN can draw on these successful experiences and improve the effect of museum floor plan generation.

(2) As can be seen from the text, the author only selected 100 data for training. This amount of data is relatively small in the field of artificial intelligence. The authors also mention that no pre-trained model was used. So how do you make sure that this amount of data is sufficient for the model to be trained?

Response: Thank you for your comments. The collection of museum floor plans is usually very costly and difficult, so it is not simple to obtain a large amount of high-quality floor plan data. Based on existing resources and practical feasibility, this study collected 100 museum floor plans from 63 cities in China. At the same time, in order to make up for the problem of insufficient data volume, this paper adopts a variety of data enhancement techniques, including rotation, flipping, cropping, and scaling. These techniques can effectively expand the diversity of training data, thereby improving the generalization ability of the model. Through data enhancement, researchers can expand 100 original datasets into thousands of variants, providing the model with richer training samples.

(3) In what proportion is the data divided into the training and test sets?

Response: Thank you for your comments. In this study, due to the limited amount of data (only 100 museum floor plans), the traditional training set and test set division was not performed. Instead, the full amount of data was used for training, and the model performance was evaluated through cross-validation and hand-drawn pattern testing. This method ensures the reliability and generalization ability of the model under limited data, effectively utilizes the information of each sample, and achieves comprehensive and reliable performance evaluation. See Section 3.2 of the paper for details.

(4) Chapter Settings and text descriptions are somewhat confusing. The article mentions "Model 1" and "Model 2" several times. Why not give them specific names to distinguish them better?

Response: Thank you for your comments. The reasons for using the names "Model 1" and "Model 2" in this study are as follows:

(a) Maintaining research consistency. Using the naming method of "Model 1" and "Model 2" is consistent with past studies and ensures research consistency. This naming method makes it easier for readers to understand the relationship between this study and previous studies, especially in the review and citation of previous work. (b) Highlighting the relationship between the previous and subsequent steps of the model. This study used CGAN to train two models, and the two models have a previous and subsequent step relationship in use: Model 1 is responsible for generating functional partitions from the outline of the museum floor plan; Model 2 generates the final museum floor plan based on the functional partitions generated by Model 1. This naming method can clearly highlight the sequential relationship between the two models, allowing readers to clearly understand the process and operation steps of the model.

(5) section 3.2 and 3.3 can be combined into one section.

Response: Thank you for your comments. We modified it.

(6) The metrics used by the authors are not commonly used in artificial intelligence. It is suggested to add artificial intelligence evaluation index of image generation performance.

Response: Thank you for your comments. We have added the following content to the text:

In this study, the researchers used multiple indicators to evaluate the performance of the model, including:(a) Evaluation of the LOSS value during training: During the model's training, we recorded and examined the loss function value (LOSS) of the generative adversarial network's generator and discriminator to gauge the effectiveness of the training and the speed of the model's convergence. (b) Pixel similarity evaluation: After the model training is completed, we objectively assess the quality of the model-generated images based on the pixel similarity between the generated and real images. (c) In addition to the objective data evaluation mentioned above, we have also incorporated a subjective user evaluation. We invite users in professional fields to subjectively evaluate the generated museum floor plan through a user questionnaire. The evaluation content encompasses the image's visual effect, the rationality of the functional division, and the overall layout's aesthetics. Rating and feedback: We score the generated images based on user feedback and collect specific opinions and suggestions.By adopting multiple objective and subjective evaluation indicators, we strive to have a comprehensive understanding of the model’s generation quality. These indicators include LOSS value during training, pixel similarity evaluation, and user evaluation. This multi-dimensional evaluation method can more comprehensively and accurately reflect the model's performance and generation effects.

(7) Who is the "other" in Figure 8? Please indicate it clearly in the drawing.

Response: Thank you for your comments. "Others" refers to the models studied by other scholars. This section is to compare the differences between past studies and this study. In order to have a clearer and more specific description, we decided to add it in the figure title.

Reviewer #2: In this work, author presented an innovative approach to using artificial intelligence, specifically the CGAN model, to enhance the space design of exhibition halls in urban cultural museums. The study addresses the limitations of traditional space design methods and highlights the potential benefits and application prospects of integrating AI into the design process. Here is the point author may want to highlight:

The static nature of trained models may limit their adaptability to evolving design trends and changing user needs. Continuous updating and retraining of the model with new data are essential to keep the generated designs relevant and up-to-date. Is it possible to make adaptable according to changed requirement? Every time training and testing with new data will increase the computational and cost both.

Response: Thank you for your comments. In response to this question, we would add the following: 

In order to enable the model to adapt to changing design trends and user needs, future research can use incremental learning and transfer learning so that the model can gradually learn new data and new features without the need for complete retraining. For example, incremental learning enables the model to gradually accumulate knowledge, while transfer learning uses pre-trained models to quickly adapt to new design trends and needs through fine-tuning. Both methods can effectively reduce the computational cost of each training session.

---

## [Decision Letter · Decision Letter 1]

3 Sep 2024

Artificial Intelligence Empowering Museum Space Layout Design: Insights from China

PONE-D-24-12619R1

Dear Dr. Chen,

We’re pleased to inform you that your manuscript has been judged scientifically suitable for publication and will be formally accepted for publication once it meets all outstanding technical requirements.

Kind regards,

Sushank Chaudhary, Ph.D

Academic Editor

PLOS ONE

Additional Editor Comments (optional):

Reviewers' comments:

Reviewer's Responses to Questions

**Comments to the Author**

1. If the authors have adequately addressed your comments raised in a previous round of review and you feel that this manuscript is now acceptable for publication, you may indicate that here to bypass the “Comments to the Author” section, enter your conflict of interest statement in the “Confidential to Editor” section, and submit your "Accept" recommendation.

Reviewer #1: All comments have been addressed

Reviewer #2: All comments have been addressed

2. Is the manuscript technically sound, and do the data support the conclusions?

Reviewer #1: Yes

Reviewer #2: Yes

3. Has the statistical analysis been performed appropriately and rigorously? 

Reviewer #1: Yes

Reviewer #2: Yes

4. Have the authors made all data underlying the findings in their manuscript fully available?

Reviewer #1: Yes

Reviewer #2: (No Response)

5. Is the manuscript presented in an intelligible fashion and written in standard English?

Reviewer #1: Yes

Reviewer #2: Yes

6. Review Comments to the Author

Reviewer #1: The authors have carefully revised all the suggestions and given reasonable explanations. I think it meets the publication requirements of the journal.

Reviewer #2: (No Response)

7. PLOS authors have the option to publish the peer review history of their article (what does this mean?). If published, this will include your full peer review and any attached files.

Reviewer #1: No

Reviewer #2: No

---

## [Editor Report · Acceptance letter]

9 Sep 2024

PONE-D-24-12619R1 

PLOS ONE

Dear Dr. Chen, 

I'm pleased to inform you that your manuscript has been deemed suitable for publication in PLOS ONE. Congratulations! Your manuscript is now being handed over to our production team.

Kind regards, 

on behalf of

Prof. Sushank Chaudhary 

Academic Editor

PLOS ONE